# Attn-QAT: 4-Bit Attention With Quantization-Aware Training

Peiyuan Zhang [* 1]   Matthew Noto [* 2]   Wenxuan Tan [* 3]   Chengquan Jiang [4]   Will Lin [1]   Wei Zhou [5]   Hao Zhang [1]

## Abstract

Achieving reliable 4-bit attention is a prerequisite for end-to-end FP4 computation on emerging FP4-capable GPUs, yet attention remains the main obstacle due to FP4's tiny dynamic range and attention's heavy-tailed activations. This paper presents the first systematic study of 4-bit quantization-aware training (QAT) for attention. We find "drop-in" QAT – naively combining an FP4 forward pass with high-precision Flash Attention (FA)-style backward pass – leads to training instability. We identify two key principles for stable FP4 attention: (1) matching low-precision recomputation of attention scores in the backward pass and (2) resolving implicit precision assumptions in FA's gradient calculation. Based on these insights, we propose Attn-QAT and implement fused Triton kernels for training and FP4 inference kernels. Across diffusion and language models, Attn-QAT recovers the quality drop from FP4 attention without explicit outlier-mitigation heuristics used in prior FP4 attention, and delivers up to a 1.5x speedup on an RTX 5090 over SageAttention3 and up to a 1.3x speedup over FA4 on a B200. Video demos can be found here. Code can be found here.

## 1. Introduction

As model sizes and deployment scales continue to grow, quantization has emerged as a key technique for reducing memory footprint and improving inference throughput. While 8-bit inference has been widely adopted in production systems (Liu et al., 2024a; Xiao et al., 2023; Kwon et al., 2023; Zhang et al., 2024b), the introduction of native FP4 Tensor Core support in NVIDIA's Blackwell architecture creates new opportunities for 4-bit quantization (Abecas-

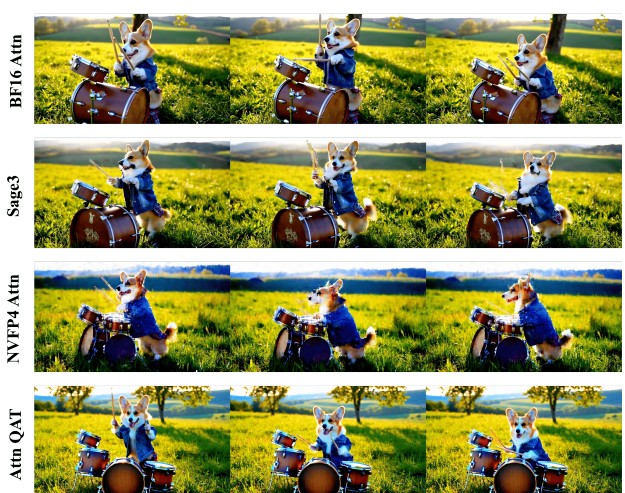

Figure 1. Both NVFP4 attention and SageAttention3 suffer from a significant quality drop on Wan 2.1 14B. Our proposed method, Attn-QAT, recovers the quality drop by using quantization-aware training. Note that temporal inconsistency is hard to visualize in sampled frames. We attach video samples in Appendix A **without cherry-picking** to better showcase the superior quality of Attn-QAT.

sis et al., 2025), offering up to a 2x increase in arithmetic intensity together with lower memory traffic. However, despite recent progress in attention quantization, state-of-the-art methods such as the SageAttention series (Zhang et al., 2024b;a; 2025) still suffer from significant quality degradation when pushed to 4-bit attention.

We trace this degradation to two intrinsic challenges in FP4 attention quantization. First, FP4 provides an extremely coarse value set and narrow dynamic range (only 15 distinct values), leaving little room for post-training calibration to preserve attention dynamics. Second, compared to linear layers, attention exhibits heavier-tailed activation distributions and more outliers, making it substantially more sensitive to numerical precision. Even with SageAttention's mitigation techniques—such as Q/K smoothing and two-level quantization—the resulting precision is still insufficient to reliably recover quality. This motivates a different approach: quantization-aware training (QAT) (Jacob et al., 2018), where model weights are updated to compensate for the errors introduced by 4-bit execution.

QAT typically simulates low-precision execution (e.g., FP4)

---

[*]Equal contribution   [1]UC San Diego  [2]Stanford University  [3]University of Wisconsin-Madison  [4]Independent Researcher  [5]Georgia Institute of Technology. Correspondence to: Hao Zhang <haozhang@ucsd.edu>.

*Proceedings of the 43rd International Conference on Machine Learning*, Seoul, South Korea. PMLR 306, 2026. Copyright 2026 by the author(s).

in the forward pass, while computing gradients in higher precision to update model weights. While this paradigm has been well explored for linear layers, to our knowledge no prior work has successfully applied QAT to attention. Modern attention implementations, such as FlashAttention (FA) (Dao et al., 2022), are realized as heavily fused operators whose backward pass relies on recomputation and precision-sensitive algebraic identities. Consequently, we find that naively switching the forward pass to FP4 while reusing FA's BF16 backward pass kernel produces exploding gradients, indicating that stable attention QAT requires careful precision coordination between the forward and backward recomputation intermediates.

In this paper, we present the first systematic study of quantization-aware training for the attention operation. Through detailed analysis, we identify two key requirements for stable Attn-QAT. First, the recomputation of the attention probability matrix $\mathbf{P}$ during the backward pass must use the same low precision as the forward pass, ensuring consistency with the intermediate activations. Second, FlashAttention relies on the identity $\mathbf{P}_i^\top \mathbf{dP}_i = \mathbf{dO}_i^\top \mathbf{O}_i$, to maintain linear memory complexity for the backward pass, which only holds when the forward and backward passes share the same precision. When the forward pass is executed in FP4 and the backward pass in BF16, this assumption breaks. To resolve this, we compute the attention output $\mathbf{O}$ in both low and high precision during the forward pass, storing the high-precision output solely for gradient computation.

We implement both forward and backward pass Triton kernels for Attn-QAT training, and improve SageAttention3 CUDA kernels for inference. Experiments on both diffusion models and large language models show Attn-QAT recovers the quality loss introduced by FP4 attention without relying on any outlier suppression mechanisms proposed in various versions of SageAttention. By eliminating these additional operations, we can achieve 1.1x-1.5x speedup over SageAttention3 on an RTX 5090. In summary, we make the following contributions:

- We conduct the first systematic study of quantization-aware training for attention, identify the key inconsistencies that arise in the attention backward pass, and propose a principled solution.

- We implement efficient custom FP4 attention kernels for both QAT training and inference.

- We demonstrate that Attn-QAT fully recovers model quality without any outlier mitigation techniques, delivering significant speedups on an RTX 5090.

**Conflict of Interest Disclosure.** The authors declare no financial conflicts of interest related to this work.

## 2. Methods

This section provides the technical background and then details of our approach. We first review NVFP4 microscaling format and the training-free SageAttention3 method, which leverages native FP4 matrix multiplication with additional heuristics for accuracy recovery. We then introduce quantization-aware training for attention and describe how Attn-QAT adapts QAT to FlashAttention-style fused operators. Finally, we detail the backward-pass modifications required for stable training and summarize our kernel-level implementation.

### 2.1. NVFP4 and SageAttention3

Microscaling FP4 (MXFP4) (Rouhani et al., 2023) is a block floating-point quantization scheme in which a tensor is partitioned into small fixed-size blocks (32). Elements within each block are stored in FP4 format and share an E8M0 scale factor. NVIDIA's NVFP4 adopts this microscaling principle with a smaller block size (16) and E4M3 scaling factors for more fine-grained scaling. Following Zhang et al. (2025), we adopt NVFP4 for all FP4 operations in this paper.

Given a tensor $X \in \mathbb{R}^{N \times d}$, NVFP4 quantization applies block-wise symmetric quantization through a microscaling operator $\phi$. The tensor is partitioned into blocks $X_{ij} \in \mathbb{R}^{1 \times 16}$, where each block shares a single scale factor $s_{ij}$. The quantization process is defined as

$$\phi(\mathbf{X}): \quad s_{ij} = \frac{\max(|\mathbf{X}_{ij}|)}{6}, \qquad \hat{\mathbf{X}}_{ij} = \left\lceil \frac{\mathbf{X}_{ij}}{s_{ij}} \right\rceil \quad (1)$$

where $\lceil \cdot \rceil$ denotes rounding to the nearest FP4-representable value. Dequantization recovers the high-precision format via

$$\phi^{-1}(\hat{\mathbf{X}}, s): \quad \mathbf{X}'_{ij} = s_{ij} \cdot \hat{\mathbf{X}}_{ij}. \quad (2)$$

Blackwell GPUs provide native support for NVFP4 matrix multiplication via a dedicated hardware primitive:

$$\mathbf{C} = \text{FP4MM}(\mathbf{A}, \hat{s}_A, \mathbf{B}, \hat{s}_B) \quad (3)$$

SageAttention3 (Zhang et al., 2025) is a training-free NVFP4 attention method that builds on the native FP4 matrix multiplication primitive in Eq. (3), and introduces additional heuristics to mitigate the accuracy degradation caused by aggressive 4-bit quantization. To reduce the impact of outliers when computing $QK^\top$, it applies smoothing to both queries and keys by subtracting block-wise means along the token dimension. Given a query block $\mathbf{Q}_i$ and a key block $\mathbf{K}_j$, smoothing is defined as

$$\gamma(\mathbf{Q}_i) = \mathbf{Q}_i - \bar{\mathbf{q}}_i,$$
$$\gamma(\mathbf{K}_j) = \mathbf{K}_j - \bar{\mathbf{k}}. \quad (4)$$

**Algorithm 1 Attn-QAT Forward (Inference)**

1: **Require** $\mathbf{Q} \in \mathbb{R}^{N_q \times d}$, $\mathbf{K}, \mathbf{V} \in \mathbb{R}^{N_k \times d}$, tile sizes $B_q, B_k$
2: **Require** NVFP4 quantizer $\phi(\cdot)$ returning $(\hat{\mathbf{X}}, \hat{\mathbf{s}}_\mathbf{X})$
3: Partition $\mathbf{Q}$ into tiles $\{\mathbf{Q}_i\}_{i=1}^{T_q}$ of size $B_q \times d$; partition $\mathbf{K}, \mathbf{V}$ into tiles $\{\mathbf{K}_j, \mathbf{V}_j\}_{j=1}^{T_k}$ of size $B_k \times d$
4: $(\hat{\mathbf{Q}}, \hat{\mathbf{s}}_\mathbf{Q}), \ (\hat{\mathbf{K}}, \hat{\mathbf{s}}_\mathbf{K}), \ (\hat{\mathbf{V}}, \hat{\mathbf{s}}_\mathbf{V}) \leftarrow \phi(\mathbf{Q}), \ \phi(\mathbf{K}), \ \phi(\mathbf{V})$
5: **for** $i = 1$ to $T_q$ **do**
6:    $\mathbf{m}_i \leftarrow -\infty, \ \mathbf{l}_i \leftarrow \mathbf{0}, \ \mathbf{O}_i \leftarrow \mathbf{0}$
7:    **for** $j = 1$ to $T_k$ **do**
8:      $\mathbf{S} \leftarrow \text{FP4MM}(\hat{\mathbf{Q}}_i, \hat{\mathbf{s}}_\mathbf{Q}, \hat{\mathbf{K}}_j, \hat{\mathbf{s}}_\mathbf{K})/\sqrt{d}$
9:      $\mathbf{m}_{\text{new}} \leftarrow \max(\mathbf{m}_i, \text{rowmax}(\mathbf{S}))$
10:     $\alpha \leftarrow \exp(\mathbf{m}_i - \mathbf{m}_{\text{new}}), \ \tilde{\mathbf{P}} \leftarrow \exp(\mathbf{S} - \mathbf{m}_{\text{new}})$
11:     $\mathbf{l}_i \leftarrow \alpha \odot \mathbf{l}_i + \text{rowsum}(\tilde{\mathbf{P}}), \ \ \mathbf{m}_i \leftarrow \mathbf{m}_{\text{new}}$
12:     $(\hat{\tilde{\mathbf{P}}}, \hat{\mathbf{s}}_{\tilde{\mathbf{P}}}) \leftarrow \phi(\tilde{\mathbf{P}})$
13:     $\mathbf{O}_i \leftarrow \text{diag}(\alpha)\mathbf{O}_i + \text{FP4MM}(\hat{\tilde{\mathbf{P}}}, \hat{\mathbf{s}}_{\tilde{\mathbf{P}}}, \hat{\mathbf{V}}_j, \hat{\mathbf{s}}_\mathbf{V})$
14:    **end for**
15:    $\mathbf{O}_i \leftarrow \text{diag}(\mathbf{l}_i)^{-1}\mathbf{O}_i, \ \ \mathbf{L}_i \leftarrow \mathbf{m}_i + \log(\mathbf{l}_i)$
16: **end for**
17: **Return** $\mathbf{O}, \mathbf{L}$

**Algorithm 2 Attn-QAT Forward (Training)**

1: **Require** $\mathbf{Q} \in \mathbb{R}^{N_q \times d}$, $\mathbf{K}, \mathbf{V} \in \mathbb{R}^{N_k \times d}$, tile sizes $B_q, B_k$
2: $\mathbf{Q}^F \leftarrow \phi^{-1}(\phi(\mathbf{Q})), \quad \mathbf{K}^F \leftarrow \phi^{-1}(\phi(\mathbf{K})), \quad \mathbf{V}^F \leftarrow \phi^{-1}(\phi(\mathbf{V}))$ {fake quantization}
3: Partition $\mathbf{Q}^F$ into tiles $\{\mathbf{Q}_i^F\}_{i=1}^{T_q}$ of size $B_q \times d$; partition $\mathbf{K}^F, \mathbf{V}^F$ into tiles $\{\mathbf{K}_j^F, \mathbf{V}_j^F\}_{j=1}^{T_k}$ of size $B_k \times d$
4: **for** $i = 1$ to $T_q$ **do**
5:    $\mathbf{m}_i \leftarrow -\infty, \ \mathbf{l}_i \leftarrow \mathbf{0}, \ \mathbf{O}_i \leftarrow \mathbf{0}, \ \mathbf{O}'_i \leftarrow \mathbf{0}$
6:    **for** $j = 1$ to $T_k$ **do**
7:      $\mathbf{S} \leftarrow \mathbf{Q}_i^F(\mathbf{K}_j^F)^\top/\sqrt{d}$
8:      $\mathbf{m}_{\text{new}} \leftarrow \max(\mathbf{m}_i, \text{rowmax}(\mathbf{S}))$
9:      $\alpha \leftarrow \exp(\mathbf{m}_i - \mathbf{m}_{\text{new}}), \ \tilde{\mathbf{P}} \leftarrow \exp(\mathbf{S} - \mathbf{m}_{\text{new}})$
10:     $\tilde{\mathbf{P}}^F \leftarrow \phi^{-1}(\phi(\tilde{\mathbf{P}}))$ {fake quantization}
11:     $\mathbf{l}_i \leftarrow \alpha \odot \mathbf{l}_i + \text{rowsum}(\tilde{\mathbf{P}}), \ \ \mathbf{m}_i \leftarrow \mathbf{m}_{\text{new}}$
12:     $\mathbf{O}_i \leftarrow \text{diag}(\alpha)\mathbf{O}_i + \tilde{\mathbf{P}}^F \mathbf{V}_j^F$
13:     $\mathbf{O}'_i \leftarrow \text{diag}(\alpha)\mathbf{O}'_i + \tilde{\mathbf{P}}\mathbf{V}_j^F$ {high-precision output for backward}
14:    **end for**
15:    $\mathbf{O}_i \leftarrow \text{diag}(\mathbf{l}_i)^{-1}\mathbf{O}_i, \ \ \mathbf{O}'_i \leftarrow \text{diag}(\mathbf{l}_i)^{-1}\mathbf{O}'_i, \ \ \mathbf{L}_i \leftarrow \mathbf{m}_i + \log(\mathbf{l}_i)$
16: **end for**
17: **Return** $\mathbf{O}, \mathbf{L}, \mathbf{O}'$

where $\bar{\mathbf{q}}_i = \text{mean}(\mathbf{Q}_i)$ and $\bar{\mathbf{k}} = \text{mean}(\mathbf{K})$ are broadcasted to all tokens in the block. With this decomposition, the attention score can be written as

$$\begin{aligned} \mathbf{S}_{ij} &= (\bar{\mathbf{q}}_i + \gamma(\mathbf{Q}_i))(\bar{\mathbf{k}} + \gamma(\mathbf{K}_j))^\top \\ &= \gamma(\mathbf{Q}_i)\gamma(\mathbf{K}_j)^\top + \Delta\mathbf{S}_{ij} + \mathbf{b}. \end{aligned} \tag{5}$$

where $\Delta\mathbf{S}_{ij} = \bar{\mathbf{q}}_i\gamma(\mathbf{K}_j)^\top$ and $\mathbf{b} = \bar{\mathbf{q}}_i\bar{\mathbf{k}}^\top + \gamma(\mathbf{Q}_i)\bar{\mathbf{k}}^\top$.

In addition, because the softmax output $\mathbf{P}$ takes values in $[0, 1]$, it does not sufficiently utilize the range of NVFP4. SageAttention3 first rescales each row of $\mathbf{P}$ to between $[0, 448 \times 6]$ (where 6 is the maximum value of **FP4e2m1** and 448 is the maximum value of the **FP8e4m3** scale factor), and then applies standard FP4 quantization, enabling more effective use of NVFP4 precision during attention computation.

## 2.2. Quantization Aware Training

Note that Eq. (3) is equivalent to:

$$\mathbf{C} = \text{BF16MM}(\phi^{-1}(\phi(\mathbf{A})), \phi^{-1}(\phi(\mathbf{B}))) \tag{6}$$

Quantization-aware training (QAT) builds upon Eq. (6) and refers to the operation $\phi^{-1}(\phi(\cdot))$ as *fake quantization*. Conceptually, this corresponds to a standard high-precision forward pass in which fake quantization is applied to the inputs of every matrix multiplication, thereby emulating Eq. (3) during training. During the backward pass, QAT relies on the straight-through estimator (STE) to approximate gradients with respect to the quantized inputs. Specifically, the backward pass still operates in high-precision with the

gradients computed as:

$$\begin{aligned} d\mathbf{A} &\approx d\big(\phi^{-1}(\phi(\mathbf{A}))\big) \\ &= \text{BF16MM}\big(d\mathbf{C}, \ \phi^{-1}(\phi(\mathbf{B}))^\top\big), \\ d\mathbf{B} &\approx d\big(\phi^{-1}(\phi(\mathbf{B}))\big) \\ &= \text{BF16MM}\big(\phi^{-1}(\phi(\mathbf{A}))^\top, \ d\mathbf{C}\big). \end{aligned} \tag{7}$$

In short, QAT only modifies a normal BF16 training loop by applying fake quantization to the inputs of matrix multiplication operations, while everything else, including the forward and backward precision, is kept the same. By explicitly optimizing the model under NVFP4 constraints, QAT updates the weights to compensate for the accuracy loss induced by low-bit quantization.

## 2.3. Attn-QAT

Attn-QAT adopts the simplest NVFP4 attention implementation, as illustrated in Algorithms 1 and 2. Rather than incorporating the outlier-mitigation heuristics proposed in Zhang et al. (2025), we rely on quantization-aware training to recover the quality loss. However, applying quantization-aware training to attention is non-trivial, as FlashAttention's tightly fused operator design limits fine-grained customization. In standard attention, there are two matrix multiplications: the score computation $\mathbf{S} = \mathbf{Q}\mathbf{K}^\top$ and the value aggregation $\mathbf{O} = \mathbf{P}\mathbf{V}$. Under QAT, these operations correspond to applying fake quantization to $\mathbf{Q}$ and $\mathbf{K}$ in the former case, and to $\mathbf{P}$ and $\mathbf{V}$ in the latter, as specified in Eq. (6) and Eq. (7). However, FlashAttention computes attention by tiling the input and recomputing activations in the

**Algorithm 3 Attn-QAT backward**

1: **Require** $\mathbf{Q}^F \in \mathbb{R}^{N_q \times d}$, $\mathbf{K}^F, \mathbf{V}^F \in \mathbb{R}^{N_k \times d}$, $\mathbf{dO} \in \mathbb{R}^{N_q \times d}$, $\mathbf{L} \in \mathbb{R}^{N_q}$, $\mathbf{O}' \in \mathbb{R}^{N_q \times d}$, tile sizes $B_q, B_k$
2: **Ensure** $\mathbf{dQ}, \mathbf{dK}, \mathbf{dV}$
3: $\mathbf{D} \leftarrow \text{rowsum}(\mathbf{dO} \odot \mathbf{O}')$ {uses high-prec $\mathbf{O}'$}
4: Partition into tiles: $\{\mathbf{Q}_i^F, \mathbf{dO}_i, \mathbf{L}_i, \mathbf{D}_i\}_{i=1}^{T_q}$ with $B_q$ rows, $\{\mathbf{K}_j^F, \mathbf{V}_j^F\}_{j=1}^{T_k}$ with $B_k$ rows
5: Initialize $\mathbf{dQ} \leftarrow \mathbf{0}, \ \mathbf{dK} \leftarrow \mathbf{0}, \ \mathbf{dV} \leftarrow \mathbf{0}$
6: **for** $j = 1$ to $T_k$ **do**
7:    $\mathbf{dK}_j \leftarrow \mathbf{0}, \ \mathbf{dV}_j \leftarrow \mathbf{0}$
8:    **for** $i = 1$ to $T_q$ **do**
9:       $\mathbf{S} \leftarrow \mathbf{Q}_i^F (\mathbf{K}_j^F)^\top / \sqrt{d}$
10:      $\mathbf{P} \leftarrow \exp(\mathbf{S} - \mathbf{L}_i)$
11:      $\mathbf{P}^F \leftarrow \phi^{-1}(\phi(\mathbf{P}))$ {recompute in same low precision as FWD}
12:      $\mathbf{dV}_j \mathrel{+}= (\mathbf{P}^F)^\top \mathbf{dO}_i$
13:      $\mathbf{dP} \leftarrow \mathbf{dO}_i (\mathbf{V}_j^F)^\top$
14:      $\mathbf{dS} \leftarrow \mathbf{P} \odot (\mathbf{dP} - \mathbf{D}_i)/\sqrt{d}$
15:      $\mathbf{dQ}_i \mathrel{+}= \mathbf{dS}\,\mathbf{K}_j^F$
16:      $\mathbf{dK}_j \mathrel{+}= \mathbf{dS}^\top \mathbf{Q}_i^F$
17:    **end for**
18:    Write $\mathbf{dK}_j, \mathbf{dV}_j$ into the corresponding tiles of $\mathbf{dK}, \mathbf{dV}$ in global memory for all $j$
19: **end for**
20: Write $\mathbf{dQ}_i$ into the corresponding tile of $\mathbf{dQ}$ in global memory for all $i$
21: **Return** $\mathbf{dQ}, \mathbf{dK}, \mathbf{dV}$

---

backward pass, leading to two subtle but critical mismatches with standard QAT.

**Matching the precision of $\mathbf{P}$ in the forward and backward passes.** In FlashAttention, the full attention probabilities $\mathbf{P}$ are not materialized nor saved in the forward pass. Instead, they are recomputed in the backward pass from the stored log-sum-exp vector $\mathbf{L}$. Under QAT, this recomputation must exactly match the numerical precision of the forward pass. To address this, Attn-QAT **explicitly fake-quantizes** the recomputed $\mathbf{P}$ in the backward pass (line 11 of Alg. 3), ensuring that gradients are computed with respect to the same low-precision activations used in the forward pass.

**High-precision $\mathbf{O}$ for the backward pass.** A second subtlety arises from the softmax backward computation in FlashAttention. Given a row-wise softmax $\mathbf{P}_i = \text{softmax}(\mathbf{S}_i)$, its Jacobian satisfies

$$\begin{aligned}
\mathbf{dS}_i &= \big(\text{diag}(\mathbf{P}_i) - \mathbf{P}_i\mathbf{P}_i^\top\big)\,\mathbf{dP}_i \\
&= \mathbf{P}_i \odot \mathbf{dP}_i - (\mathbf{P}_i^\top\mathbf{dP}_i)\,\mathbf{P}_i.
\end{aligned} \tag{8}$$

Note that, like in standard attention implementations, the softmax operates in FP32 precision to avoid numerical instability (even in FP4 attention), so we use the high-precision FP32 activation $\mathbf{P}$ instead of $\mathbf{P}^F$ to compute the $\mathbf{dS}_i$ term.

The scalar term $\mathbf{P}_i^\top \mathbf{dP}_i$ requires access to the full row of attention probabilities, which results in quadratic memory complexity in the sequence length. To achieve linear memory complexity in the backward pass, we follow FlashAttention and exploit the identity

$$\begin{aligned}
\mathbf{P}_i^\top \mathbf{dP}_i &= \sum_j \mathbf{P}_{ij}\,\mathbf{dO}_i^\top \mathbf{V}_j^F \\
&= \mathbf{dO}_i^\top \sum_j \mathbf{P}_{ij}\mathbf{V}_j^F \\
&= \mathbf{dO}_i^\top \mathbf{O}_i'.
\end{aligned} \tag{9}$$

The first equality relies on $\mathbf{dP}_i = \mathbf{dO}_i^\top \mathbf{V}_j^F$, which is easy to derive by plugging in Eq. (7). The last equality relies on $\mathbf{O}_i = \sum_j \mathbf{P}_{ij}\mathbf{V}_j^F$. However, the output tile $\mathbf{O}_i$ is computed during the forward pass as

$$\mathbf{O}_i = \sum_j \mathbf{P}_{ij}^F \mathbf{V}_j^F,$$

meaning that the identity in Eq. (9) no longer holds if $\mathbf{O}_i$ is used directly. Thus, to preserve the correctness of the backward computation, we must additionally calculate a high-precision output tile

$$\mathbf{O}_i' = \sum_j \mathbf{P}_{ij}\mathbf{V}_j^F$$

during the forward pass, with the full high-precision matrix $\mathbf{O}'$ being used exclusively to compute the scalar term $\mathbf{dO}_i^\top\mathbf{O}_i' = \mathbf{P}_i^\top\mathbf{dP}_i$ in the backward pass.

### 2.4. Training and RTX 5090 Inference Kernel

We implement our training kernels by extending the Triton reference attention kernel (Tillet et al., 2019) and inserting fake quantization at the appropriate locations, as specified in Algorithm 2 and Algorithm 3. For quantization and dequantization between high-precision formats and NVFP4, we leverage inline PTX on Blackwell GPUs using the new `cvt.rn.satfinite.e2m1x2.f32` and `cvt.rn.f16x2.e2m1x2` instructions. On non-Blackwell GPUs, we instead implement NVFP4 emulation via explicit bitwise operations. This design allows our training kernels to run on any NVIDIA GPU supported by Triton, while still exploiting native NVFP4 instructions when available.

To fully realize the performance benefits of FP4 attention during inference, we use custom CUDA kernels rather than Triton. Our inference kernel is adapted from SageAttention3's CUDA implementation with minor modifications. We use this CUDA kernel during inference for diffusion models. For language model evaluation, we modify the Triton paged-attention implementation in vLLM (Kwon et al., 2023) to support NVFP4 fake quantization.

## 2.5. B200 Inference Kernel

To support datacenter Blackwell GPUs (B200), we implement a CuTe-DSL (NVIDIA, 2025) FP4 attention kernel extending FlashAttention-4 (FA4) (Dao et al., 2026; 2025). The kernel matches the Attn-QAT forward path and additionally supports mixing block-scaled NVFP4/MXFP8 and FP8 `tcgen05.mma` for $\mathbf{QK}^\top$ and $\mathbf{PV}$ GEMMs. We briefly explain the implementation techniques below.

**Bypassing the softmax bottleneck on B200.** As FP4 Tensor Cores deliver up to $4\times$ the arithmetic intensity of BF16 on standalone GEMMs, one might naturally expect a comparable end-to-end win for FP4 attention. We found this is surprisingly not the case on B200. Block-scaled $\mathbf{QK}$ yields at most $1.3\times$ speedup over BF16 FA4 (Figure 5b), and moreover, **additionally quantizing $\mathbf{PV}$ is slower than pure BF16 forward** (Appendix §B.2). The root cause is asymmetric hardware scaling as mentioned in FA4 (Dao et al., 2026): B200 roughly doubles MMA throughput over H100, but leaves `MUFU.EX2` (exp) throughput unchanged—so attention shifts from GEMM-bound to jointly softmax-and-GEMM-bound, and the $4\times$ Tensor Core advantage is masked. Quantizing $\mathbf{P}$ on the fly requires computing fine-grained block scales in the softmax warp groups, which lengthens the kernel's critical softmax path. Therefore, only block-scaled $\mathbf{QK}$ with BF16 or FP8 $\mathbf{PV}$ brings speedups on B200. As B300 doubles `exp` throughput and Rubin quadruples it via FP16 `exp`, block-scaled $\mathbf{PV}$ should obtain the expected speedup on them, and we are actively working on B300 tuning. Appendix B.2 explains the $\mathbf{PV}$ quantization slowdown in detail.

**TMEM overlap schedule.** B200's `tcgen05.mma` instructions accumulate results in tensor memory (TMEM) to alleviate register pressure as tile sizes grow. However, FA4 already occupies all available TMEM with two $\mathbf{QK}$ score tiles (S1 & S2) and two partial-output tiles (O1 & O2); block-scaled $\mathbf{QK}$ and $\mathbf{PV}$ each require two scale factors in TMEM ($\mathbf{sf}_{qk1}$, $\mathbf{sf}_{qk2}$ and $\mathbf{sf}_{pv1}$, $\mathbf{sf}_{pv2}$). To reuse storage while minimizing pipeline stalls, we time-multiplex them with a **TMEM overlap schedule**: reuse S2 for $\mathbf{sf}_{qk1}$ and S1 for $\mathbf{sf}_{qk2}$, and load $\mathbf{sf}_{pv1}$ and $\mathbf{sf}_{pv2}$ only after the corresponding $\mathbf{QK}$ GEMM completes. See Appendix B for the full schedule and stall-avoidance details.

## 3. Experiments

### 3.1. Setup

**Models and Baselines.** We apply Attn-QAT to both video diffusion models and large language models. For diffusion models, we evaluate on Wan-2.1 (Wang et al., 2025a) at two scales: 1.3B and 14B. For language modeling, we evaluate on Qwen3-14B (Yang et al., 2025) and Llama 3.1-

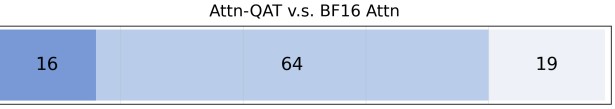

*Figure 2.* Win–Tie–Lose blind human evaluation on 99 randomly sampled VBench prompts for Wan 2.1 14B. Attn-QAT matches BF16 attention in perceived visual quality.

70B (Grattafiori et al., 2024). We compare Attn-QAT against the following attention variants: (i) BF16 attention, (ii) NVFP4 attention without training, (iii) SageAttention3, which incorporates advanced outlier mitigation techniques for FP4 attention. We exclude SageAttention3 from all LLM experiments because its open-source kernel implementation exhibits significant numerical errors in causal attention, resulting in degraded accuracy. Note that all non-attention components remain in high precision.

**Training and Evaluation Details.** For diffusion models, we generate synthetic latents using Wan-2.1-14B to perform Attn-QAT. For our experiments on Wan-2.1-1.3B, we use a dataset of 81K examples with 480P resolution. For experiments on Wan-2.1-14B, we use 13K examples with 720P resolution. We evaluate all subcategories of video quality in VBench (Huang et al., 2024), using Qwen2.5-3B-Instruct for prompt augmentation, following the guide specified in the VBench GitHub repository. Additionally, we conduct blind human evaluation on 99 randomly selected prompts from VBench.

For language models, we apply Attn-QAT as a continued pretraining procedure on base models using the C4 dataset (Raffel et al., 2020), and evaluate whether Attn-QAT can recover the quality degradation introduced by FP4. We report results on WikiText (Merity et al., 2016), HellaSwag (Zellers et al., 2019), PIQA (Bisk et al., 2020), WinoGrande (Sakaguchi et al., 2021), and ARC-C (Clark et al., 2018) using lm-eval-harness (Gao et al., 2024). We further perform supervised fine-tuning on Dolci-Instruct (Olmo et al., 2025) with both BF16 attention and Attn-QAT to verify that Attn-QAT has the same fine-tuning quality as BF16 attention. We then evaluate the fine-tuned model on a more challenging benchmark suite including MMLU-Redux (Gema et al., 2025), GPQA-Diamond (Rein et al., 2024), MATH-500 (Hendrycks et al., 2021), GSM8K (Cobbe et al., 2021), and IFEval (Zhou et al., 2023), using EvalScope (Team, 2024) and vLLM (Kwon et al., 2023). Full training configurations and training overhead breakdown are provided in the appendix.

### 3.2. Diffusion Experiments

**Main results.** Table 1 reports results on Wan 2.1 14B and Table 2 shows results on Wan 2.1 1.3B. Replacing BF16 attention with FP4 attention *without training* results in a

*Table 1.* VBench evaluation on Wan 2.1 14B. Experiments 1–3 are training-free inference baselines, while Experiment 4 applies Attn-QAT and requires additional training.

| Exp. | Wan 2.1 14B | Imaging Quality | Aesthetic Quality | Subject Consistency | Background Consistency | Temporal Flickering | Motion Smoothness | Dynamic Degree | Overall Quality |
|---|---|---|---|---|---|---|---|---|---|
| 1 | BF16 | 0.6869 | 0.6692 | 0.9572 | 0.9635 | 0.9759 | 0.9878 | 0.5193 | 0.8335 |
| 2 | FP4 | 0.6324 | 0.6271 | 0.9412 | 0.9548 | 0.9783 | 0.9855 | 0.2983 | 0.7968 |
| 3 | SageAttention3 | 0.6604 | 0.6510 | 0.9517 | 0.9584 | 0.9758 | 0.9862 | 0.4751 | 0.8203 |
| 4 | **Attn-QAT** | 0.6745 | 0.6712 | 0.9685 | 0.9716 | 0.9828 | 0.9902 | 0.3646 | 0.8279 |

*Table 2.* VBench evaluation on Wan 2.1 1.3B. Experiments 1–3 are training-free inference baselines, while Experiments 4–8 apply Attn-QAT and require additional training. Attn-QAT recovers the quality loss introduced by FP4 attention without explicit outlier mitigation techniques.

| Exp. | Wan 2.1 1.3B | Imaging Quality | Aesthetic Quality | Subject Consistency | Background Consistency | Temporal Flickering | Motion Smoothness | Dynamic Degree | Overall Quality |
|---|---|---|---|---|---|---|---|---|---|
| 1 | BF16 | 0.6728 | 0.6657 | 0.9647 | 0.9646 | 0.9832 | 0.9897 | 0.3923 | 0.8267 |
| 2 | FP4 | 0.5592 | 0.6109 | 0.9601 | 0.9605 | 0.9854 | 0.9892 | 0.1160 | 0.7785 |
| 3 | SageAttention3 | 0.5507 | 0.6163 | 0.9583 | 0.9582 | 0.9836 | 0.9886 | 0.2099 | 0.7834 |
| 4 | Attn-QAT | 0.6775 | 0.6764 | 0.9709 | 0.9706 | 0.9839 | 0.9902 | 0.3039 | 0.8252 |
| 5 | + SmoothK | 0.6738 | 0.6699 | 0.9664 | 0.9676 | 0.9811 | 0.9887 | 0.3425 | 0.8232 |
| 6 | + Two-level quant P | 0.6801 | 0.6782 | 0.9749 | 0.9749 | 0.9867 | 0.9918 | 0.2541 | 0.8257 |
| 7 | – High prec. O in BWD | 0.5660 | 0.4373 | 0.8709 | 0.9384 | 0.9761 | 0.9827 | 0.0331 | 0.7185 |
| 8 | – Fake quantization of P in BWD | 0.6837 | 0.6798 | 0.9727 | 0.9729 | 0.9851 | 0.9912 | 0.2652 | 0.8254 |

substantial drop across VBench metrics, as shown by the comparison between Exp. 1 and 2. While SageAttention3 partially mitigates this degradation, it still underperforms the BF16 baseline, indicating that post-training quantization alone is insufficient for FP4 attention. In contrast, Attn-QAT recovers the quality loss caused by FP4 attention, matches BF16 performance across metrics, and outperforms SageAttention3. These results demonstrate that quantization-aware training alone is sufficient to compensate for FP4 attention errors, without requiring the additional outlier-mitigation heuristics used in SageAttention3. In Figure 2, we report additional human evaluation on 99 prompts sampled from VBench, where raters find Attn-QAT outputs comparable to the BF16 baseline. Qualitative comparisons are provided in Appendix A.

**Outlier mitigation is unnecessary with Attn-QAT.** SageAttention3 differs from Attn-QAT in the forward pass in two key aspects: (i) it applies QK smoothing to increase the precision in calculating $\mathbf{S}$, and (ii) it adopts a two-level quantization scheme for the attention probability matrix $\mathbf{P}$. To isolate the effect of these design choices in training, we explicitly incorporate K smoothing [1] and two-level $\mathbf{P}$ quantization into Attn-QAT and evaluate their impact in Exp. 4–6 of Table 2. Across all evaluated metrics, we observe that introducing either K smoothing or two-level quantization yields only marginal changes compared to the vanilla Attn-QAT baseline. In particular, none of these

heuristics consistently improves performance across all evaluation dimensions, and qualitative inspection of the generated videos reveals no noticeable differences. This suggests that Attn-QAT already learns to recover from quantization error during training, rendering additional mitigation strategies largely redundant. Specifically, our analysis suggests that QAT does not significantly change the distribution of attention logits. Thus, we hypothesize the model becomes more robust to quantization noise, reducing sensitivity to outliers.

**Correct backward design is essential for stable training.** We ablate the two central design choices that enable stable Attn-QAT training. First, removing the high-precision output $\mathbf{O}'$ and instead using the low-precision $\mathbf{O}$ in the backward pass leads to severe training instability. As shown in plots (a) and (b) of Figure 3, this modification causes exploding gradients and substantially higher training loss. Consistently, Exp. 7 in Table 2 exhibits a significant drop in VBench scores. Second, omitting fake quantization of $\mathbf{P}$ during backward recomputation results in a similar final VBench score (Exp. 4 vs. Exp. 8) and comparable training loss (Figure 3, plot (b)). However, as shown in plot (a) of Figure 3, this setting produces significantly noisier gradient norms, indicating reduced training stability. These results suggest that fake quantization of $\mathbf{P}$, while not strictly required for convergence in our setup, plays an important role in stabilizing training dynamics. Finally, a naive baseline that performs an FP4 forward pass while reusing FlashAttention's BF16 backward kernel consistently results in ex-

---

[1]We skip ablating QAT with smoothing Q because it leads to complicated gradient computation.

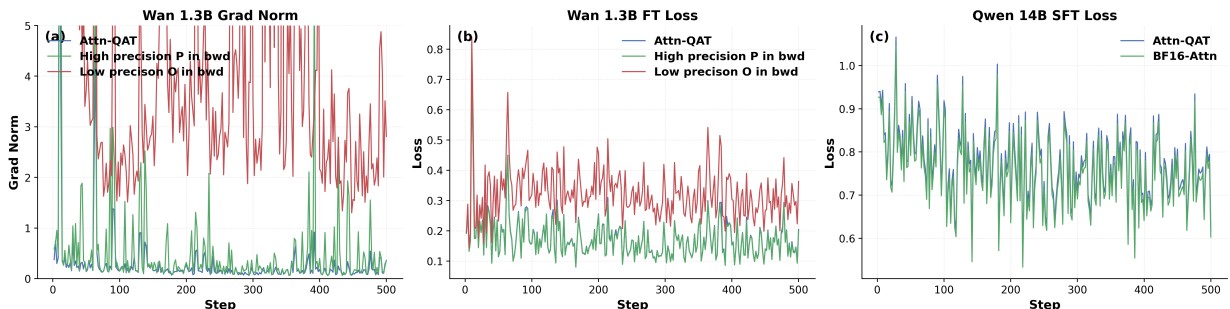

Figure 3. Training dynamics for diffusion and language models. (a–b) Gradient norm and loss during Wan 2.1 1.3B finetuning under different Attn-QAT configurations. (c) Finetuning loss curves of Qwen3-14B comparing BF16 attention and Attn-QAT.

ploding gradients; we therefore omit it from Table 2.

### 3.3. LLM Experiments

**Continued pretraining.** In Table 4, we start from the base Qwen3-14B and Llama 3.1-70B models and continue training them on the C4 dataset to evaluate whether Attn-QAT can recover the quality loss introduced by 4-bit attention. Consistent with our diffusion results, applying NVFP4 attention without training leads to clear performance degradation across all benchmarks compared to BF16 attention. In contrast, Attn-QAT recovers most of this loss. For Qwen3-14B, Attn-QAT restores performance to near-BF16 levels and even improves WinoGrande and ARC-c accuracy. For Llama 3.1-70B, Attn-QAT partially recovers the degradation but does not fully match BF16 performance. We attribute this gap primarily to limited training budget and lack of hyperparameter tuning for 70B due to hardware constraints (Appendix C.2), suggesting that longer training may further close the gap.

**Supervised fine-tuning.** To evaluate whether Attn-QAT can be applied directly during supervised fine-tuning (SFT), without requiring a separate quantization-aware training stage, we fine-tune the base models of Qwen3-14B and Llama 3.1-70B on Dolci-Instruct using either Attn-QAT or standard BF16 attention. Figure 3 (c) reports the training loss, while Table 3 summarizes downstream benchmark performance. Although Attn-QAT incurs a slightly higher training loss than BF16, it achieves nearly identical benchmark performance for Qwen3-14B across all evaluated tasks. For Llama 3.1-70B, FP4 Attn-QAT remains close to BF16 with a small gap. These results indicate that Attn-QAT can be applied as a drop-in replacement for BF16 attention during SFT, simplifying the training pipeline by removing the need for a dedicated QAT stage prior to SFT.

### 3.4. Kernel Benchmarks

Quantization-aware training can potentially introduce a train–test mismatch, since FP4 behavior is emulated via fake quantization in BF16 during training (fake quant), while in-

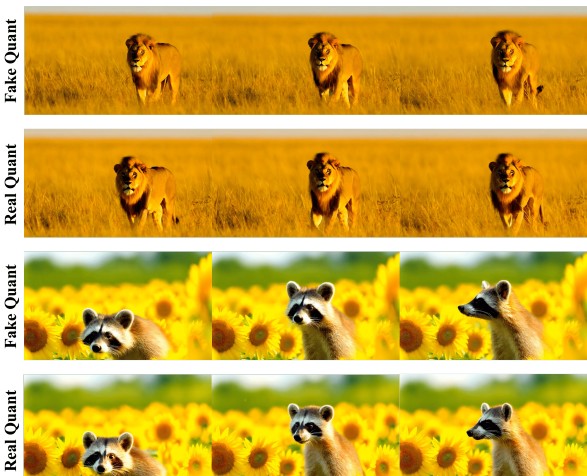

Figure 4. The Triton forward pass (fake quantization with BF16 GEMM and FP4 emulation) and the CUDA forward pass (real FP4 quantization and FP4 GEMM) produce visually indistinguishable videos, indicating close numerical agreement between the two implementations.

ference uses a real FP4-quantized GEMM (real quant). To verify that this mismatch does not occur in practice, we perform inference on identical prompts using both the forward pass of our Triton training kernel and the CUDA inference kernel. As shown in Figure 4, the two implementations produce nearly identical outputs.

We benchmark the throughput of our CUDA kernel on an RTX 5090 in Figure 5a, comparing against FlashAttention2 and SageAttention3. By eliminating the additional Smooth-QK and two-level quantization of $\mathbf{P}$, Attn-QAT achieves approximately 1.1x-1.5x higher throughput than SageAttention3. We attribute this speedup primarily to the reduced preprocessing overhead for $\mathbf{Q}$ and $\mathbf{K}$[2]. We do not observe additional measurable kernel-level speedup from simplifying $\mathbf{P}$ quantization.

---

[2]Our evaluation setup slightly differs from SageAttention3 in that we include the latency of input preprocessing (smoothing and quantization).

*Table 3.* LLM Finetuning Results

| Exp | Model | Precision | MMLU-Redux | IFeval | GPQA-Diamond | MATH-500 | GSM8K |
|-----|-------|-----------|------------|--------|--------------|----------|-------|
| 1 | Qwen3-14B | BF16 | 0.8316 | 0.7107 | 0.4495 | 0.8060 | 0.9295 |
| 2 | | FP4 w. Attn-QAT | 0.8392 | 0.7306 | 0.4394 | 0.7840 | 0.9098 |
| 3 | Llama 3.1-70B | BF16 | 0.7928 | 0.8637 | 0.4091 | 0.5300 | 0.8840 |
| 4 | | FP4 w. Attn-QAT | 0.7823 | 0.8532 | 0.3838 | 0.5120 | 0.8673 |

*Table 4.* Benchmark results for LLM continued pretraining.

| Exp. | Model | Precision | MMLU | WinoGrande | ARC-c | HellaSwag | PIQA | WikiText↓ |
|------|-------|-----------|------|------------|-------|-----------|------|-----------|
| 1 | Qwen3-14B | BF16 | 0.8044 | 0.7403 | 0.5922 | 0.8140 | 0.8215 | 0.5700 |
| 2 | | FP4 | 0.7965 | 0.7214 | 0.5734 | 0.8050 | 0.8052 | 0.5763 |
| 3 | | Attn-QAT | 0.7984 | 0.7585 | 0.6084 | 0.8034 | 0.8188 | 0.5778 |
| 4 | Llama 3.1-70B | BF16 | 0.7881 | 0.8161 | 0.6135 | 0.8575 | 0.8422 | 0.2838 |
| 5 | | FP4 | 0.7577 | 0.7656 | 0.6015 | 0.8463 | 0.8308 | 0.3275 |
| 6 | | Attn-QAT | 0.7773 | 0.7940 | 0.6153 | 0.8557 | 0.8351 | 0.3076 |

**B200 benchmarks.** Figure 5b reports B200 throughput from our CuTe-DSL FP4 kernel, which mixes block-scaled NVFP4 or MXFP8 **QK** with BF16 or FP8 **PV**. Peak NVFP4+FP8 throughput is **2018 TFLOPS** at $(b=1, s=32768, h=24, d=128)$, 1.3x over BF16 FA4. At the Wan 2.1-T2V-1.3B latent shape $(1, 32768, 12, 128)$ we measure 1913 vs. 1508 TFLOPS (1.27x). MXFP8+FP8 peaks at 1948 TFLOPS–while theoretically cutting **QK** GEMM by half or to 1/4 does not matter when softmax is the bottleneck, we found that NVFP4 **QK** is still faster due to reducing the number of issued instructions by half in the $K$ dimension[3]. Appendix B.3 gives the full precision table vs. BF16.

## 4. Related Work

**Post-Training Quantization.** Post-training quantization (PTQ) applies quantization to model weights and/or activations after a model has been fully trained. While PTQ may involve a lightweight calibration step to estimate the quantization statistics, it does not update model parameters. Early work on PTQ primarily focused on convolutional and linear layers (Wang et al., 2019; Nagel et al., 2019; Xiao et al., 2023; Lin et al., 2024), with recent efforts extending these techniques to attention operators, most notably in the SageAttention series (Zhang et al., 2024b;a; 2025). A central challenge in PTQ is the presence of activation and weight outliers, which can induce large quantization errors under low-bit representations. As a result, most recent PTQ methods emphasize outlier suppression. SmoothQuant addresses activation outliers by migrating quantization dif-

ficulty from activations to weights through a reparameterization (Xiao et al., 2023). SageAttention (Zhang et al., 2025) introduces attention-specific techniques, including Q/K smoothing and two-level quantization of the attention probabilities. These methods achieve near-lossless performance at 8-bit precision; however, empirical evidence shows that their accuracy degrades under more aggressive 4-bit settings, particularly for attention quantization.

**Quantization-Aware Training.** Quantization-aware training (QAT) introduces a lightweight training phase after full-precision training and before deployment. QAT incorporates quantization effects during training by simulating low-precision arithmetic in the forward pass while using higher-precision gradients for optimization, typically via straight-through estimators (Bengio et al., 2013; Yin et al., 2019). QAT has been successfully applied to convolutional and fully connected layers, enabling robust deployment under low-bit constraints (Gong et al., 2019; Jacob et al., 2018; Liu et al., 2024b). To our knowledge, prior work has not systematically studied QAT for the attention operation itself. This is important because modern large-scale models rely on fused attention kernels such as FlashAttention (Dao et al., 2022), where materializing the full attention matrix is impractical. Therefore, practical quantized attention must work within this fused paradigm. The instability we study arises from the interaction between quantization and fused attention's recomputation-based backward pass, making it critical for real-world deployment.

**Native Low-Bit Training.** Native low-bit training differs fundamentally from QAT by executing low-precision matrix multiplication in both the forward and backward passes. By performing all major computations in low precision, native low-bit training can improve not only inference ef-

---

[3]Each `tcgen05.mma` instruction always processes 256 bits along $K$, regardless of dtype and $M/N$ tile shape.

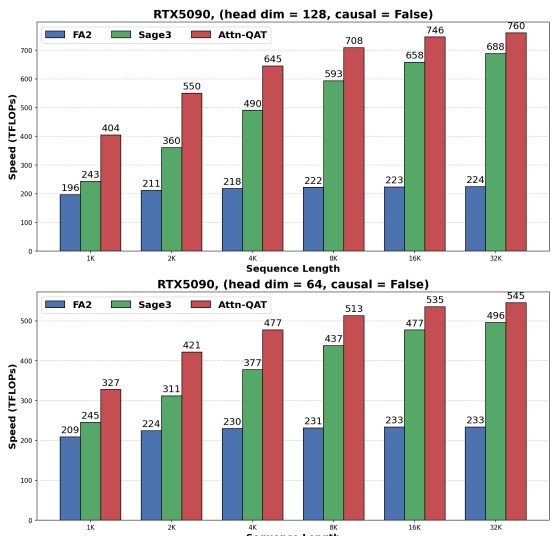

*(a)* Kernel throughput on RTX 5090. We compare attention kernel performance with head dimensions 128 (top) and 64 (bottom), using a batch size of 16 and 16 attention heads. All results report end-to-end throughput.

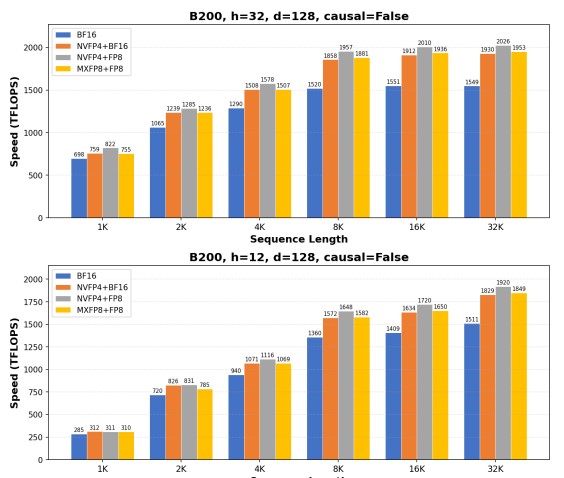

*(b)* B200 attention throughput (TFLOPS). NVFP4 + BF16 means NVFP4 QK + BF16 PV GEMMs

ficiency but also training throughput, and is therefore typically used to train models from scratch (Peng et al., 2023; Fishman et al., 2024; Hernández-Cano et al., 2025). For example, DeepSeek-V3 demonstrates the feasibility of training a frontier-scale model using naive 8-bit linear layers (Liu et al., 2024a), and recent work has begun to explore native 4-bit training for linear operators (Abecassis et al., 2025; Wang et al., 2024; 2025b; Chmiel et al., 2025). Progress on native low-bit training for attention remains limited. To our knowledge, SageAttention3 (Zhang et al., 2025) is the first work that explores native 8-bit training for attention, and there are no prior studies investigating native 4-bit training for attention mechanisms.

# 5. Conclusion & Future Work

We introduce Attn-QAT, the first systematic study of 4-bit quantization-aware training for attention. We show that naively applying QAT to FP4 attention fails due to precision mismatches in the backward pass, and identify two requirements for stability: low-precision recomputation of attention probabilities and a high-precision auxiliary output for correct softmax gradients. With these improvements, Attn-QAT enables stable training of FP4 attention. Experiments on diffusion models and large language models show that Attn-QAT recovers BF16-level quality without any outlier mitigation heuristics, demonstrating that QAT alone is sufficient for reliable 4-bit attention.

Our current Attn-QAT implementation supports both RTX 5090 and B200 GPUs; the B200 kernel supports block-sparse attention but not paged attention, which requires storing the scale factors in page table format. Additionally, we are still working on B300 PV quantization tuning, which will enable full FP4 GEMM speedups for attention in datacenter-scale serving. Finally, we expect to integrate 4-bit KV caches into a mainstream serving library to enable full low-precision decoding and further reduce memory overhead during inference. We have open-sourced our RTX 5090 kernel and B200 kernel and will continue developing them.

## Impact Statement

Our work targets efficient serving of foundation models by developing low-bit attention kernels that substantially increase throughput without sacrificing output quality. By lowering the computational cost, our approach makes high-quality text/video generation more accessible to researchers and practitioners with constrained hardware resources, thereby broadening the applicability of generative AI in domains such as creative production and education. Although increased generation speed may raise concerns about potential misuse, existing safeguards—including content detection and watermarking techniques—provide practical mechanisms for risk mitigation. Overall, our method contributes to more efficient and lower-carbon serving systems, while misuse risks should be addressed through complementary safeguards such as content provenance, detection, and watermarking.

## Acknowledgements

The work is supported by UCSD HDSI, Nvidia, and a faculty research award from Google. The computing resources were provided by MBZUAI IFM and Nvidia's donation.

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

# A. More Qualitative Results

We provide additional qualitative comparisons in Figure 6, Figure 7, and Figure 8, and **include more demos here without cherry-picking**. The results show that Attn-QAT produces substantially higher-quality videos than SageAttention3 and achieves visual quality comparable to BF16 attention.

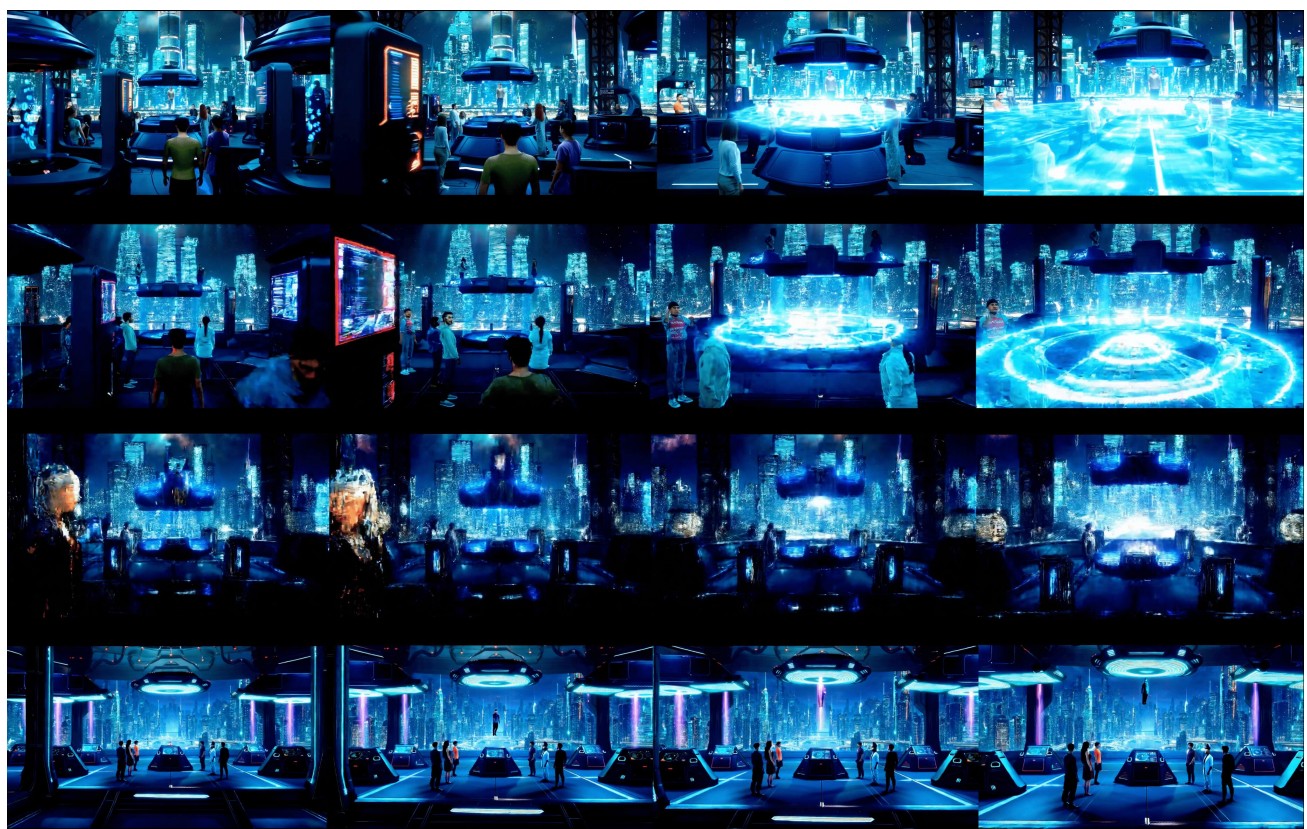

*Figure 6. In a futuristic world where teleportation technology has become a reality, a bustling cityscape filled with towering skyscrapers and advanced architecture stands in the background. Amidst this backdrop, a group of diverse individuals, each with unique appearances and expressions, gather around a central chamber equipped with shimmering teleportation devices. The scene captures various stages of teleportation – from individuals floating mid-air before vanishing, to others appearing instantly in their destinations. The lighting is dramatic, with neon lights flickering and casting shadows across the faces of the teleportees. The camera moves between subjects, capturing moments of awe and excitement as they teleport, emphasizing the rapidity and efficiency of the new technology. The futuristic cityscape provides a vivid contrast to the serene yet chaotic scene within the teleportation chamber. Cinematic and high-tech visual style, focusing on the emotional impact of teleportation on the characters. Medium shot and wide shots showcasing the teleportation process.*

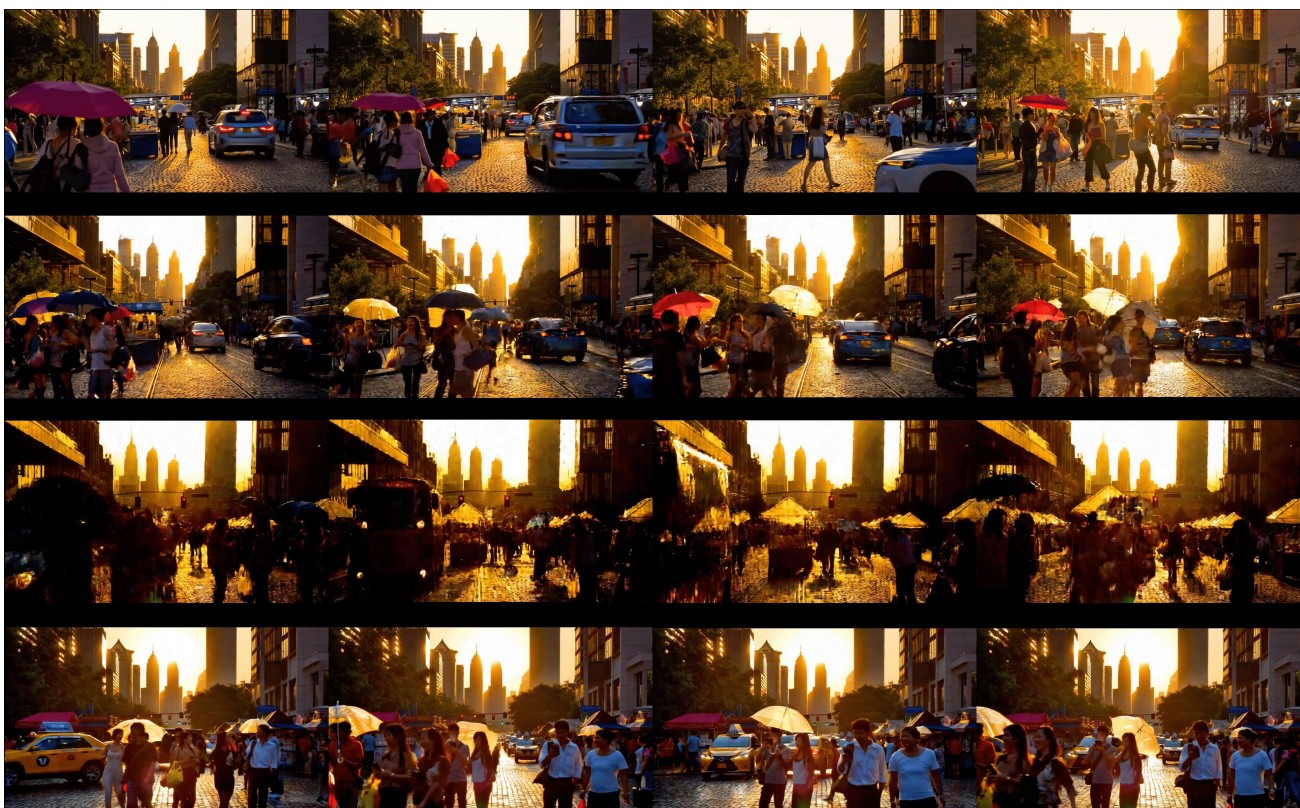

*Figure 7. Downtown street scene captured in a vibrant sunset, bustling with activity. A diverse crowd of people walk down the cobblestone streets, carrying bags and umbrellas. Cars honk and taxis weave through the narrow alleys. Street vendors set up their stalls, offering snacks and drinks. A group of friends laugh and chat as they take pictures together. The backdrop is a picturesque downtown skyline, with towering skyscrapers and modern architecture reflecting the golden hues of the setting sun. People are seen walking with various expressions, some looking at their phones, others lost in thought. The scene captures the energy and excitement of a lively downtown area. City lights start to flicker as the sun sets lower in the sky. The entire scene is filled with natural motion, with people moving about, vehicles driving, and the sun slowly descending. Downtown night-time atmosphere with warm lighting and soft shadows. Medium shot of the bustling street, full-body shots of people interacting, and low-angle shots of the skyline.*

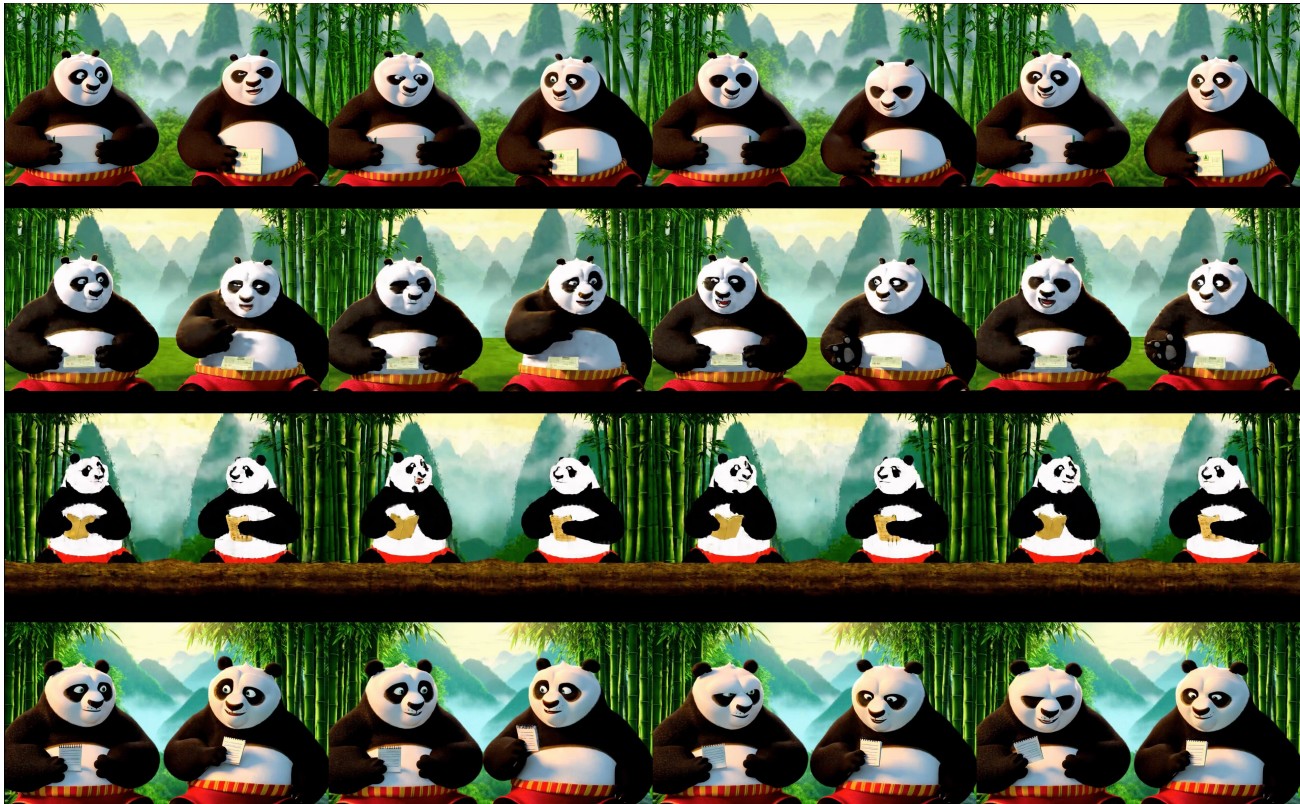

*Figure 8. CG animation digital art, two adorable pandas sitting side-by-side on a bamboo forest backdrop. The pandas have expressive faces, one looking thoughtful with a raised eyebrow, the other with a curious look. They are both wearing traditional panda costumes with bright red sashes tied around their waists. Each panda holds a small notebook in front of them, depicting an academic paper. The background features lush bamboo forests and misty mountain peaks. The pandas are engaged in animated conversation, occasionally pointing at their notes. Soft lighting casts a warm glow over the scene. Detailed digital artwork with realistic textures. Low-angle view, medium shot side-by-side seating.*

## B. B200 FP4 Attention Kernel

This appendix supplements §2.5 with additional implementation and measurement detail for our Blackwell (B200) inference kernel (Zhang et al., 2026), implemented in CuTe-DSL atop FlashAttention-4 (Dao et al., 2026; 2025).

**Block-scaled MMA and tensor memory.** Blackwell exposes native block-scaled FP4/FP8 GEMMs via `tcgen05.mma.cta_group.kind.block_scale`, applying per-group dequantization scales inside the Tensor Cores. Unlike Hopper `wgmma`, MMA accumulators live in tensor memory (TMEM, 128 lanes × 512 columns per SM), so attention must copy scores TMEM→registers for softmax and registers→TMEM before the **PV** GEMM. FA4 overlaps **QK** MMA and softmax across warp groups (Dao et al., 2026), but on B200 the softmax `MUFU.EX2` throughput is unchanged from H100 while MMA throughput roughly doubles, so long-context attention remains softmax-bound.

### B.1. TMEM overlap schedule

On B200, scale factors must be loaded global memory (GMEM)→shared memory (SMEM) (via TMA)→TMEM and duplicated across the four warps in a warp group via a `tcgen05.cp` multicast before `tcgen05.mma` can consume them. With 128 × 128 tile sizes, however, FA4's pipeline **already uses all available TMEM**: S1 and S2 hold **QK** outputs (128 columns each), and O1/O2 use the remaining 256 columns (Figure 9).

Our TMEM overlap schedule reuses S2 for `sf_qk1` and S1 for `sf_qk2`. Because `tcgen05.mma` ops of the same shape issued by a thread are guaranteed to execute sequentially[4], we only insert barriers between S1 T2R and the `sf_qk2` load.

---

[4]See NVIDIA PTX `tcgen05` pipelined-instruction memory consistency.

`sf_vp1` is scheduled after $\mathbf{QK}_2$ and therefore cannot stomp on S1; `sf_vp2` uses a barrier to wait for S2 copy-out (this barrier rarely fires because there are two MMAs between $\mathbf{QK}_2$ and $\mathbf{PV}_2$). We also considered storing $\mathbf{P}$ in SMEM to free TMEM, but rejected it due to insufficient `ldmatrix` shapes (max $8 \times 8$ vs. $32 \times 16$ for `tcgen05.st`).

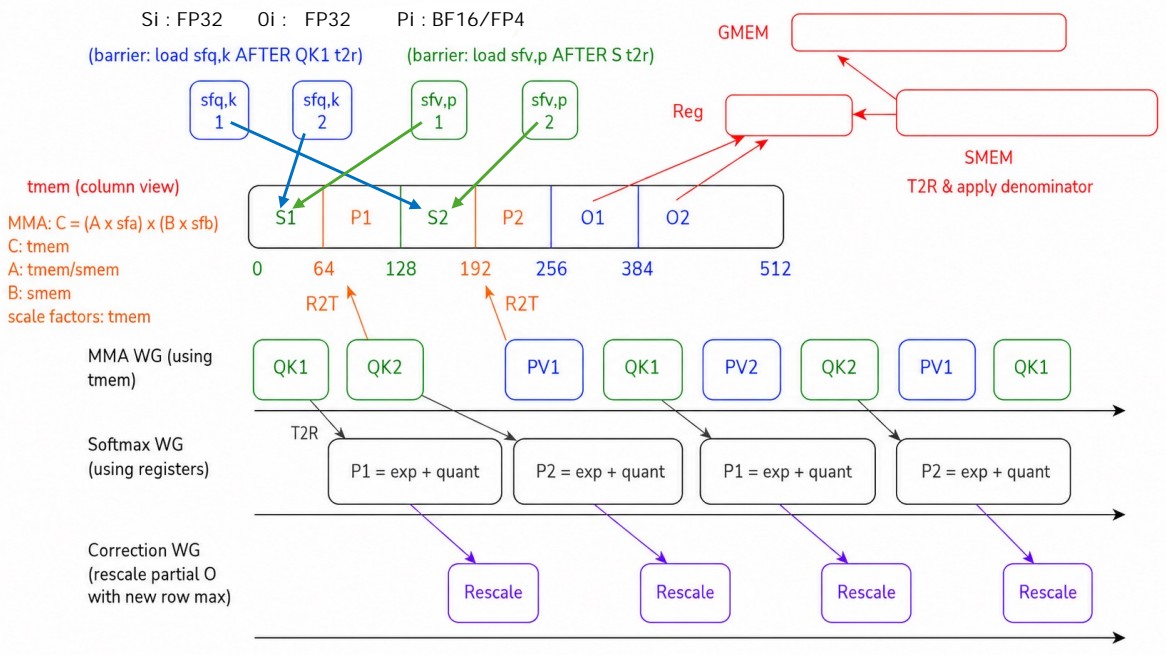

*Figure 9.* **TMEM overlap schedule** for block-scaled $\mathbf{QK}$ on B200 ($128 \times 128$ tiles). S1/S2 hold $\mathbf{QK}$ scores (128 TMEM columns each); O1/O2 hold partial outputs (256 columns). We time-multiplex scale-factor buffers (`sf_qk1`, `sf_qk2`) on S2/S1 between score T2R and the next MMA, with a barrier before loading `sf_qk2`; $\mathbf{PV}$ scale loads are scheduled after the corresponding $\mathbf{QK}$ output is copied out.

## B.2. Softmax bottleneck and block-scaled $\mathbf{PV}$

From Table 5, B200 has the same `MUFU.EX2` ops/clk/SM as H100 but roughly doubles MMA throughput, and B300 doubles `MUFU.EX2` ops/clk/SM at the cost of increasing TDP by 400W. Other resources such as registers, shared memory, and CUDA cores roughly remain the same, which creates the softmax bottleneck: at tile size $128 \times 128$, softmax takes the same number of cycles as two BF16 GEMMs, and FA4 (Dao et al., 2026) overlaps their execution by placing them into separate warp groups (see Figure 9) of four warps each. FP4 GEMMs only moderately improve the overall runtime by improving the imperfect overlap between MMA and softmax warp groups caused by the pipeline warmup phase (issuing two $\mathbf{QK}$ GEMMs) and other miscellaneous instructions. When quantizing $\mathbf{PV}$ on the fly, the group-wise scale factor computation for $\mathbf{P}$, `cvt.rn.satfinite` dtype casting instructions, and scale-factor registers→SMEM→TMEM copies sits on the softmax warp's critical path, making it slower than pure BF16 attention. While FA4 proposes softmax emulation to replace `exp` with polynomial approximation, applying it to more than 25% of the tiles causes significant register spilling and competes for CUDA Core resources with the scale factor computation, so it does not resolve the slowdown on B200.

Table 6 reports NVFP4 block-scaled $\mathbf{PV}$ latency (on-the-fly $\mathbf{P}$ quant + block-scaled $\mathbf{PV}$) vs. BF16 $\mathbf{PV}$. At long context the kernel is slower than BF16 despite higher MMA throughput, confirming the softmax-bound regression above.

*Table 5.* GPU hardware evolution relevant to attention (Tensor Core vs. softmax scaling).

| Spec | A100 (SXM4) | H100 (SXM5) | B200 (HGX) | B300 / GB300 | Rubin |
|------|-------------|-------------|------------|--------------|-------|
| Architecture | Ampere | Hopper | Blackwell | Blackwell Ultra | Rubin |
| Year | 2020 | 2022 | 2024 | 2025 | 2026 |
| Die config | 1 die, 826 mm$^2$ | 1 die, 814 mm$^2$ | 2×800 mm$^2$ | 2×800 mm$^2$ | 2× reticle + 2 I/O |
| Transistors | 54.2B | 80B | 208B | 208B | 336B |
| TDP | 400W | 700W | 1,000W | 1,100W / 1,400W | ∼1,800W |
| SMs (enabled) | 108 | 132 | 148 | 160 | 224 |
| CUDA cores (FP32) | 6,912 | 16,896 | 18,944 | 20,480 | TBD |
| BF16 Tensor TFLOPS (dense) | 312 | 989 | 2,250 | ∼2,500 | 4 PFLOPS[‡] |
| FP8 Tensor TFLOPS (dense) | — | 1,979 | 4,500 | 5,000 | 17.5 PFLOPS[‡] |
| FP4 Tensor PFLOPS (dense) | — | — | 9.0 | 14–15 | 50/35 PFLOPS[‡] |
| Registers/SM | 64K×32b | 64K×32b | 64K×32b | 64K×32b | 64K×32b |
| TMEM/SM | — | — | 256 KB | 256 KB | 256 KB+ |
| Shared mem/SM (max) | 164 KB | 228 KB | 228 KB | 228 KB | TBD |
| MUFU.EX2 ops/clk/SM | **16** | **16** | **16** | **32** | **32/64**[†] |

[†]Rubin MUFU.EX2 ops/clk/SM from NVIDIA Vera Rubin platform blog. Entries marked TBD are not yet published.

[‡]Rubin BF16, FP8, and NVFP4 dense Tensor throughputs (per GPU) from NVIDIA Vera Rubin NVL72 specifications. NVFP4 50/35 PFLOPS are peak inference and training, respectively.

*Table 6.* B200 NVFP4 block-scaled $\mathbf{PV}$ latency vs. BF16 $\mathbf{PV}$.

| Config $(b, s, h, d)$ | NVFP4 $\mathbf{PV}$ (ms) | BF16 $\mathbf{PV}$ (ms) | Ratio |
|------|------|------|------|
| $(1, 256, 16, 128)$ | 0.028 | 0.039 | 1.42× |
| $(1, 1024, 16, 128)$ | 0.027 | 0.041 | 1.52× |
| $(4, 4096, 16, 128)$ | 1.287 | 1.217 | 0.95× |
| $(1, 4096, 12, 128)$ | 0.435 | 0.336 | 0.77× |
| $(1, 32768, 12, 128)$ | 13.693 | 12.775 | 0.93× |
| $(1, 4096, 24, 128)$ | 0.538 | 0.486 | 0.90× |
| $(1, 32768, 24, 128)$ | 27.053 | 22.617 | 0.84× |

## B.3. Numerical error vs. BF16 reference

Table 7 reports cosine similarity / max abs. diff / mean abs. diff vs. BF16 flash_attn_func (NVFP4: FlashInfer block scales; MXFP8: torch FP8 + E8M0). All entries have cosine similarity around 0.99.

*Table 7.* Per-call error vs. BF16 reference on B200 (cos / max / mean).

| Config $(b, s, h, d)$ | NVFP4+BF16 | NVFP4+FP8 | MXFP8+FP8 |
|------|------|------|------|
| $(1, 256, 16, 128)$ | .9910/.1562/.0106 | .9904/.1475/.0109 | .9986/.0605/.0042 |
| $(1, 1024, 16, 128)$ | .9908/.1846/.0055 | .9901/.2119/.0057 | .9986/.0459/.0022 |
| $(4, 4096, 16, 128)$ | .9906/.0445/.0028 | .9899/.0432/.0029 | .9985/.0215/.0011 |
| $(1, 32768, 16, 128)$ | .9904/.0112/.0010 | .9897/.0122/.0010 | .9985/.0057/.0004 |
| $(4, 4096, 32, 128)$ | .9905/.0605/.0028 | .9898/.0713/.0029 | .9985/.0225/.0011 |
| $(1, 4096, 12, 128)$ | .9906/.0674/.0028 | .9899/.0771/.0029 | .9985/.0146/.0011 |
| $(1, 32768, 12, 128)$ | .9903/.0175/.0010 | .9896/.0194/.0010 | .9985/.0042/.0004 |
| $(1, 4096, 24, 128)$ | .9905/.0586/.0028 | .9898/.0645/.0029 | .9985/.0215/.0011 |
| $(1, 32768, 24, 128)$ | .9905/.0115/.0010 | .9899/.0142/.0010 | .9985/.0046/.0004 |
| $(1, 32768, 24, 64)$ | .9899/.0215/.0010 | .9892/.0223/.0011 | N/A[*] |

[*]Not supported yet by the CuTe-DSL TMA code generator due to <4 tiles in the atom_k dimension ($d$=64).

## C. Detailed Training Setup

### C.1. Diffusion Experiments

All major training jobs for Wan-2.1-1.3B were conducted on a GB200 NVL72 and used 16 B200s. We use bf16 mixed-precision training with a global batch size of 16, 16 data-parallel groups for efficient batch processing, the AdamW optimizer ($\beta_1 = 0.9, \beta_2 = 0.999$) with a learning rate of $1 \times 10^{-6}$, weight decay factor of 0.01, and the standard rectified flow matching loss as our objective. We trained these models for 4000 steps (which took roughly 12.5 hours) but noticed that the quality of our generated validation videos was better at around 3000 steps, so we opted to use the 3000 step checkpoint for inference. Because Attn-QAT requires keeping around extra buffers for the fake quantized Q, K, V, and high-precision O tensors, we needed to use full gradient checkpointing to avoid running into OOM errors.

Attn-QAT introduces additional cost only during the post-training QAT stage. In the forward pass, it requires computing an additional high-precision attention output $O'$, which increases the forward FLOPs by roughly 50%. The backward pass has the same asymptotic computation as standard attention training, and inference introduces no additional FLOPs because only the low-precision output $O$ is used. During training, the main memory overhead comes from storing the additional high-precision $O'$ together with the fake-quantized $Q$, $K$, and $V$ buffers, resulting in about 25% attention-level memory overhead. In practice, this led to a wall-clock training slowdown of approximately $1.2\times$–$2\times$ across experiments. Since Attn-QAT is applied only as a short post-training stage, this overhead is modest compared with full model pretraining; for example, our Wan-2.1-1.3B QAT run took roughly 12.5 hours, while the resulting inference efficiency gains persist throughout deployment.

We trained Wan-2.1-14B on 64 H200s (8 nodes of 8 H200s in the shared cluster we used). The mixed-precision policy, optimizer, weight decay factor, and loss are the same as for the 1.3B model experiments except we now use HSDP (Hybrid Sharded Data Parallel) with a replication dimension of 8 (across nodes) and a sharding dimension of 8 (within a node). Initially we wanted to try and use a global batch size of 64 but to avoid OOM errors, we needed to also use Ulysses with 2 sequence parallel groups to reduce the memory required to store activations. Thus, we ended up using a global batch size of 32. We finetuned the 14B model for 400 training steps which took 1 day.

For our preliminary experiments of finetuning Wan-2.1-1.3B using SageAttention3 with a naive BF16 backward pass, we used 4 RTX 5090s with 16 gradient accumulation steps, utilizing both Ulysses sequence parallelism and data parallelism across the machines. This resulted in OOM errors at around step 200 during the first validation stage. Note that all of the other hyperparameters are the same as the rest of our Wan-2.1-1.3B training jobs as explained above.

### C.2. LLM Experiments

Due to resource constraints, we did not perform any hyperparameter tuning for LLM experiments, and the largest run takes almost 6 hours on 4 B200 GPUs.

**Continued pretraining.** To study whether Attn-QAT can recover the quality degradation introduced by FP4 attention, we perform continued pretraining on base language models using the C4 dataset (Raffel et al., 2020). We conduct experiments on Qwen3-14B and Llama 3.1-70B, using the English subset of C4 and training on a 10% shard of the dataset.

All continued pretraining experiments are run on 4 NVIDIA B200 GPUs with bf16 mixed-precision. For Qwen3-14B, we use a maximum sequence length of 8192, a per-device batch size of 4, and train for up to 2000 optimization steps. For Llama 3.1-70B, due to higher memory requirements, we use a per-device batch size of 1 with gradient accumulation of 2 and train for 4000 steps. We adopt the AdamW optimizer with a learning rate of $5 \times 10^{-6}$ and enable activation checkpointing for all runs; for the 70B model, we additionally shard the token embedding and output layers to reduce memory usage.

We compare BF16 attention, naive FP4 attention without training, and FP4 attention trained with Attn-QAT under identical optimization settings. Model quality is evaluated using lm-eval-harness (Gao et al., 2024) on WikiText (Merity et al., 2016), HellaSwag (Zellers et al., 2019), PIQA (Bisk et al., 2020), WinoGrande (Sakaguchi et al., 2021), and ARC-C (Clark et al., 2018).

**Supervised fine-tuning.** To evaluate whether Attn-QAT can be used as a drop-in replacement for BF16 attention during supervised fine-tuning, we fine-tune Qwen3-14B and Llama 3.1-70B base models on the Dolci-Instruct-SFT dataset (Olmo et al., 2025). All SFT experiments are conducted on 4 NVIDIA B200 GPUs using bf16 mixed-precision training.

For Qwen3-14B, we use a maximum sequence length of 8192, a per-device batch size of 8 with gradient accumulation of 4, resulting in a global batch size of 128 tokens per step. For Llama 3.1-70B, due to higher memory requirements, we use a sequence length of 4096, a per-device batch size of 2, and the same gradient accumulation factor of 4. Both models are trained for a single epoch with a maximum of 2000 optimization steps. We adopt the AdamW optimizer with a learning rate of $5 \times 10^{-6}$ and enable activation checkpointing for all experiments; activation offloading is additionally enabled for the 70B model to avoid out-of-memory errors.

