# OpenReview forum: "Attn-QAT: 4-Bit Attention With Quantization-Aware Training"
_ICML.cc/2026/Conference — ICML 2026 regular_

### Official Review · Reviewer_3hrY · 2026-03-08

**Soundness:** 4
**Presentation:** 3
**Significance:** 4
**Originality:** 3
**Overall Recommendation:** 5
**Confidence:** 3

**Summary:**

This paper introduces Attn-QAT, a systematic study on 4-bit (NVFP4) Quantization-Aware Training (QAT) for attention. When recomputing the attention scores during the Flash Attention backward pass, Attn-QAT employs the same low precision to prevent forward-backward mismatch. Furthermore, since the identity used for gradient computation in Flash Attention relies on high-precision outputs, Attn-QAT maintains an additional high-precision output during the forward pass specifically for gradient computation. Unlike SageAttention, Attn-QAT does not require explicit heuristic methods for outliers. Experimental results demonstrate that Attn-QAT can recover the quality loss introduced by FP4 in both video diffusion models and Large Language Models, while achieving significant speedups.

**Compliance With Llm Reviewing Policy:**

Affirmed.

**Final Justification:**

The authors' rebuttal has addressed my concerns regarding overhead and scope. I keep my score and hope the authors will include all important discussions in the revised manuscript.

**Key Questions For Authors:**

1. In Algorithm 2, maintaining an additional high-precision $O'$ increases GPU memory usage. Could the authors please clarify this memory increase in actual training? Does it limit the batch size?
2. Considering that in the paper's experiments all non-attention components remain in high precision, could the authors provide the QAT training cost for a fully quantized network and discuss under what resource conditions choosing Attn-QAT would yield better returns?

**Limitations:**

No other limitations.

**Strengths And Weaknesses:**

**Strengths**

1. Attention mechanisms are typically a bottleneck in quantization. The motivation of this paper is well-founded, the research topic is novel, and the work holds high practical value.
2. The paper provides an in-depth analysis of the specific characteristics of Flash Attention in the context of QAT (specifically regarding recomputation and precision mismatch), which effectively supports the novelty of Attn-QAT.
3. In the experiment section, the paper provides additional analytical conclusions regarding quantized attention, including the non-necessity of outlier mitigation and the correct backward design, demonstrating the effectiveness and advantages of the Attn-QAT method.
4. The paper implements the attention operator for NVFP4. The acceleration effects and practicality of Attn-QAT are convincing, and the engineering implementation appears reproducible.
5. The experiments in the paper are sufficient, with validation conducted in two mainstream domains (video generation and language models), providing a comprehensive comparison.

**Weaknesses**

1. Although QAT can deliver better inference performance, it introduces additional training overhead. Could the authors provide a detailed breakdown of the training costs associated with Attn-QAT?
2. The research scope of this paper is somewhat limited, as the problems addressed by Attn-QAT are strongly correlated with FlashAttention. Can this method offer a more general perspective on improving quantized attention mechanisms?

---

> ### Author Rebuttal · Authors · 2026-03-31
>
> We thank the reviewer for the positive evaluation and thoughtful questions. We address the concerns below.
>
> > W1 Training Overhead
>
> We agree that a detailed breakdown is important and will include it in the revision (see also response to Reviewer Qcve W1). In brief, Attn-QAT introduces ~50% additional forward FLOPs during training, ~25% attention-level memory overhead (from storing O′), and results in a 1.2×–2× training slowdown. Importantly, this cost is incurred only during a short post-training stage, while the inference efficiency gains persist throughout deployment.
>
>
> > W2 Limited Scope
>
> Fused attention (e.g., FlashAttention) is the standard implementation in modern large-scale models, as materializing the full attention matrix is prohibitively expensive. As a result, any practical quantized attention method must operate within this fused computation paradigm. The instability we identify arises precisely from the interaction between quantization and fused attention’s backward design, and therefore addressing this issue is necessary for any real-world deployment of quantized attention, not specific to a particular implementation.
>
> > Q1
>
> Attn-QAT introduces only one additional activation, O′. This corresponds to approximately 25% overhead at the attention module level, and is significantly smaller at the full transformer level when accounting for MLP and other components. In practice, this overhead does not significantly limit batch size..
>
> > Q2
>
> Our preliminary results suggest that Attn-QAT is fully compatible with linear layer quantization, and can be seamlessly combined with existing low-precision training or post-training quantization pipelines.
>
> As discussed in our response to Reviewer Qcve (W1), Attn-QAT introduces only a lightweight post-training cost compared to pretraining, while delivering significant reductions in serving cost on FP4-capable hardware such as Blackwell GPUs. Therefore, we believe that for models intended for real-world deployment on such hardware, applying Attn-QAT is always a practical and beneficial step.

---

> > ### Author Rebuttal · Reviewer_3hrY · 2026-04-01
> >
> > The authors' rebuttal has addressed my concerns in detail and provided supporting data. I have no further questions. I hope the authors will include all important discussions (overhead, extended scope) in the revised manuscript.

---

### Official Review · Reviewer_Qcve · 2026-03-10

**Soundness:** 2
**Presentation:** 4
**Significance:** 4
**Originality:** 2
**Overall Recommendation:** 5
**Confidence:** 4

**Summary:**

This paper proposes Attn-QAT, a quantization-aware training method designed to enable stable FP4 attention computation. The authors observe that FP4 attention suffers from severe quality degradation due to the extremely limited dynamic range of FP4 and the heavy-tailed activation distributions that commonly occur in attention operations. They further show that naively applying QAT to FlashAttention-style implementations leads to training instability, primarily due to precision mismatches between the forward and backward passes. To address this issue, the paper proposes two key modifications. First, the attention probabilities are recomputed in the backward pass using the same low-precision representation used in the forward pass to maintain numerical consistency. Second, the forward pass additionally computes a high-precision attention output that is stored solely for gradient computation in the backward pass. The authors implement custom Triton kernels for training and CUDA kernels for inference. Experiments on diffusion models (Wan-2.1 1.3B) and large language models (Qwen3-14B and Llama3.1-70B) demonstrate that Attn-QAT is able to largely recover the quality degradation caused by FP4 attention. The method also removes the need for the outlier suppression heuristics used in SageAttention and achieves a reported inference speedup of approximately 1.1×–1.5× on RTX 5090 GPUs.

**Compliance With Llm Reviewing Policy:**

Affirmed.

**Final Justification:**

The paper presents a practical approach to stable quantization-aware training for attention with strong empirical results. My initial concerns on training overhead, inference speed attribution, and novelty have been sufficiently addressed in the rebuttal.

In particular, the authors clearly quantified the training overhead, which resolves the main soundness concern and shows the cost is manageable in a short post-training stage. They also clarified that inference speedup mainly comes from removing preprocessing, improving the accuracy of the claims.

Overall, the rebuttal adequately addressed my key concerns and increased my confidence in the work, leading me to raise my score from 4 to 5.

**Key Questions For Authors:**

## **Questions**

1. What is the exact training overhead introduced by Attn-QAT in terms of FLOPs, GPU memory usage, and wall-clock training time?

2. Why does quantization-aware training eliminate the need for outlier suppression heuristics? Specifically, does QAT change the distribution of attention logits or reduce the magnitude and frequency of activation outliers?

3. How much of the reported inference speedup comes from removing outlier suppression mechanisms in SageAttention, as opposed to improvements in the core attention kernel?

4. How does Attn-QAT compare to alternative approaches such as native low-bit training or performing backward recomputation entirely in low precision?

5. How does the proposed method scale to longer sequence lengths and larger models, where the additional stored activations may become more expensive?

---
Overall, the work is technically solid and well engineered, and the empirical results are promising. However, the contribution appears to emphasize practical system design rather than introducing a fundamentally new quantization formulation. I am open to updating my score based on the authors’ responses to the concerns raised above.

**Limitations:**

yes

**Strengths And Weaknesses:**

## **Strengths**

1. The paper provides a clear and insightful analysis of why naive quantization-aware training fails when applied to FlashAttention-style implementations. In particular, it identifies precision mismatches between the forward pass and the recomputation used in the backward pass as a key source of instability. The discussion of how FlashAttention relies on specific algebraic identities during the backward pass is technically sound and highlights an underexplored challenge in applying QAT to attention.

2. Another strength of the work is that the proposed method is compatible with practical fused attention implementations. Rather than introducing a completely new attention formulation, the authors modify the computation in a way that remains compatible with existing FlashAttention-style kernels. This improves the practical applicability of the approach.

3. The paper also evaluates the method on both diffusion models and large language models, demonstrating that Attn-QAT can recover most of the performance degradation introduced by FP4 attention. In addition, the authors provide kernel-level implementations and report improvements in inference throughput, which adds practical value to the work.
---
## **Weaknesses**

1. First, the paper does not provide a sufficiently detailed analysis of the training overhead introduced by Attn-QAT. The proposed method requires computing both low-precision and high-precision attention outputs during the forward pass and storing additional tensors for backward computation. This likely increases both computational cost and memory usage, yet the paper does not clearly quantify the additional FLOPs, memory overhead, or the resulting training slowdown.

2. Second, the proposed solution appears closer to a practical engineering workaround than a fundamentally new quantization algorithm. The precision mismatch issue is addressed by introducing additional computation and storage rather than by redesigning the underlying quantization scheme itself.

3. Third, the claim that outlier suppression heuristics become unnecessary when using Attn-QAT is not fully explained. While the empirical results suggest that the method works without the heuristics used in SageAttention, the paper does not provide a clear analysis of how QAT affects the distribution of attention activations or mitigates the impact of outliers.

4. Fourth, the reported inference speedup may partly arise from the removal of preprocessing steps required by SageAttention, rather than from improvements to the core attention kernel. A careful ablation study isolating the impact of preprocessing would help clarify the source of the performance gains.

---

> ### Author Rebuttal · Authors · 2026-03-31
>
> We thank reviewer Qcve for the careful reading and constructive feedback. We especially appreciate the recognition of (1) our analysis of precision mismatch in FlashAttention backward, (2) the compatibility of our method with practical fused attention implementations, and (3) the strong empirical results across both diffusion and language models.
> We address the reviewer’s concerns below.
>
>
> > W1 Training Overhead
>
> | Aspect | Overhead / Result | Notes |
> |--------|------------------|-------|
> | **FLOPs (Forward)** | ~+50% | Due to computing additional high-precision \(O'\) |
> | **FLOPs (Backward)** | No overhead | Same complexity as standard training |
> | **FLOPs (Inference)** | No overhead | Only low-precision O is used |
> | **Memory (Training)** | ~+25% | Additional storage of high-precision \(O'\) on top of QKVO |
> | **Memory (Inference)** | No overhead | No extra tensors required |
> | **Wall-clock Training** | 1.2×–2× slower | Observed across LLM and diffusion experiments |
>
> We agree training overhead was not clearly quantified and will include it in the revision. We emphasize that this cost is modest: QAT is a short post-training stage (e.g., ~12.5 hours for Wan 1.3B), which is negligible compared to pretraining, while the inference efficiency gains persist throughout deployment.
>
> > W2 Engineering Workaround
>
> We respectfully disagree that the method is merely an engineering workaround. The instability we address is caused by a violation of implicit algebraic assumptions in FlashAttention backward, rather than by the quantization scheme itself. Our solution enforces precision consistency and preserves key identities required for correct gradient computation. To our knowledge, this is the first work that enables stable QAT for attention, which prior work has not achieved.
>
> > W3 Attention Distribution
>
> Our preliminary analysis suggests that QAT does not significantly change the distribution of attention logits. Thus, we hypothesize the model becomes more robust to quantization noise, reducing sensitivity to outliers. This explains why explicit heuristics such as smoothing or two-level quantization become unnecessary after training. We will clarify this point in the paper.
>
>
> > W4 Inference Speed
>
> The reviewer is correct that the observed inference speedup all comes from removing preprocessing steps (e.g., Q/K smoothing). While removing two-level quant P reduces instruction complexity, we do not observe additional measurable kernel-level speedup from this alone. We will make this attribution clearer.
>
>
> > Q1
>
> See response to W1.
>
> > Q2
>
> See response to W2.
>
> > Q3
>
> See response to W3.
>
> > Q4  Native Low-bit Training
>
> To our knowledge, no prior work has explored fully naive 4-bit training for attention, where both forward and backward are performed in 4-bit. In our preliminary experiments, this setting consistently leads to training divergence.
>
> > Q5 Longer Sequence & Larger Models
>
> Attn-QAT introduces only one additional activation (O′), which scales linearly with sequence length and does not incur quadratic memory growth. In practice, O′ can be offloaded to CPU memory and prefetched layer-wise (e.g., via FSDP offloading), so the peak GPU memory overhead is limited to approximately one layer’s high-precision output, making it practical even for long-context training.
>
> We also empirically validate scalability along both dimensions. For long sequences, we include experiments on Wan 2.1 14B at 720P resolution (Table 1), which involves significantly longer effective sequence lengths. For large models, we evaluate on Llama 3.1 70B (Tables 3 and 4). In both cases, Attn-QAT remains stable and effective, demonstrating that the method scales well in realistic large-scale settings.

---

> > ### Author Rebuttal · Reviewer_Qcve · 2026-04-01
> >
> > The authors have adequately addressed the concerns I previously raised. Accordingly, I am revising my score from 4 to 5.

---

### Official Review · Reviewer_5hyu · 2026-03-10

**Soundness:** 3
**Presentation:** 3
**Significance:** 3
**Originality:** 3
**Overall Recommendation:** 5
**Confidence:** 4

**Summary:**

This paper studies the challenge of enabling reliable 4-bit attention for large language models, which is a key step toward end-to-end FP4 computation on emerging hardware. The authors observe that naive quantization-aware training for attention leads to training instability due to the heavy-tailed distribution of attention activations and the limited dynamic range of FP4. To address this issue, the paper proposes Attn-QAT, a training framework specifically designed for stable 4-bit attention quantization. Experiments show that the proposed approach enables stable training and achieves competitive performance compared to higher-precision attention implementations.

**Compliance With Llm Reviewing Policy:**

Affirmed.

**Final Justification:**

The authors’ response has addressed my concerns, and I am inclined to support the acceptance of this paper.

**Key Questions For Authors:**

Please refer to the weaknesses above.

**Limitations:**

yes

**Strengths And Weaknesses:**

Strengths

- The paper targets a key bottleneck in ultra-low precision LLM inference. Enabling reliable 4-bit attention is important for achieving fully FP4 computation on future hardware.

- The paper provides a systematic investigation of quantization-aware training for attention, including training stability and design choices.

- The proposed Attn-QAT framework provides a practical recipe for stabilizing 4-bit attention training, which may be useful for practitioners working on low-precision LLM deployment.

- The paper is well organized and the motivation, problem formulation, and experimental setup are relatively easy to follow.

Weaknesses:

- The proposed quantization algorithm leans more towards engineering heuristics/tricks, and the theoretical novelty appears somewhat limited. The conceptual novelty compared with prior quantization work appears somewhat limited.

- While the paper identifies training instability caused by heavy-tailed attention activations, the theoretical or mechanistic explanation remains relatively limited.

---

> ### Author Rebuttal · Authors · 2026-03-31
>
> We thank reviewer 5hyu  for the  thoughtful feedback. We especially appreciate the recognition of the importance of enabling reliable 4-bit attention, as well as our systematic study of attention QAT and its practical value for low-precision deployment.
>
> We address the concerns on novelty and theoretical understanding below.
>
> > W1 Theoretical Novelty
>
> While our method is practically motivated, we also have non-tricks contribution:
>
> With mathematical analysis, we identify a **previously unrecognized precision inconsistency in FlashAttention backward when applying QAT (FP4 forward + BF16 backward)*, which leads to training instability (e.g., exploding gradients)**.
> We show that this breaks implicit assumptions in the gradient derivation (e.g., Eq. 9 ), and derive necessary conditions for correctness.
>
> Based on this, we propose a principled fix: precision-matched recomputation and modified backward design.
> To our knowledge, this is the first work that **enables stable QAT for attention, *which prior work has not achieved**.
>
> > W2 Mechanistic Explanation
>
> We clarify that heavy-tailed attention activations are not a new observation. Prior works have:
>
> 1. Identified heavy-tailed behavior in attention distributions [1] [2]
> 2. Studied its impact on quantization and outlier sensitivity [3] [4]
>
> **Our goal is not to re-derive these known properties or provide theoretical ground**, but to build on them and address a more practical question: How can we make 4-bit attention actually work in training and deployment without relying on heuristics?
> In contrast to prior work that relies on outlier-mitigation tricks (e.g., smoothing, multi-level quantization), we show that:
>
> 1. QAT alone can learn to compensate for heavy-tailed distributions.
>
> 2. These heuristics become unnecessary after training (Table 2)
>
> This positions our work as a practical advancement that simplifies the design space, rather than a theoretical study of activation distributions.
>
> References:
>
> [1] Sparse VideoGen: Accelerating Video Diffusion Transformers with Spatial-Temporal Sparsity
>
> [2] Fast Video Generation with Sliding Tile Attention
>
> [3] SageAttention2: Efficient Attention with Thorough Outlier Smoothing and Per-thread INT4 Quantization
>
> [4] SageAttention3: Microscaling FP4 Attention for Inference and An Exploration of 8-bit Training

---

> > ### Author Rebuttal · Reviewer_5hyu · 2026-04-07
> >
> > The authors’ response has addressed my concerns, and I am inclined to support the acceptance of this paper.

---

### Official Review · Reviewer_K66S · 2026-03-13

**Soundness:** 3
**Presentation:** 3
**Significance:** 3
**Originality:** 3
**Overall Recommendation:** 4
**Confidence:** 4

**Summary:**

This paper studies quantization-aware training (QAT) for FP4 attention, aiming to enable stable end-to-end 4-bit attention computation on FP4-capable GPUs. The authors show that naively applying QAT to attention leads to training instability due to a mismatch between the low-precision simulation in the forward pass and the higher-precision recomputation used in the backward pass.

To address this issue, the paper proposes Attn-QAT, which (1) recomputes the attention scores $\mathbf{P}$ in low precision during the backward pass to match the forward computation, and (2) additionally computes and stores the attention output $\mathbf{O}$ in high precision to enable stable gradient computation.

Experiments on diffusion and language models demonstrate that Attn-QAT substantially recovers the quality degradation observed in prior FP4 attention methods, while also providing practical speed improvements on FP4-capable GPUs.

**Compliance With Llm Reviewing Policy:**

Affirmed.

**Key Questions For Authors:**

- Figure 3. (a) shows that keeping $\mathbf{O}$ in low precision leads to exploding gradients. Could the authors provide a theoretical explanation for this phenomenon?
- The ablation results indicate that keeping $\mathbf{P}$ in low precision during the backward pass yields only limited performance gains. Could the authors clarify the necessity of this design choice?
- The authors attribute the speedup over SageAttention-3 to reduced preprocessing overhead for $\mathbf{Q}$ and $\mathbf{K}$. It is suggested to include a more detailed comparison covering both quantization time and kernel execution, to better clarify the source of the observed speedup.

**Limitations:**

yes

**Strengths And Weaknesses:**

### Strengths

1. The paper provides a useful analysis of why naive QAT fails for attention, identifying precision mismatch between forward simulation and backward recomputation as a key source of instability. This insight could be valuable for future research on low-precision training.
2. The paper provides an efficient implementation with CUDA and triton kernels, enabling practical deployment and noticeable speed improvements on FP4-capable hardware.
3. The paper presents evaluations on both diffusion models and language models, demonstrating the effectiveness of the proposed approach.

### Weaknesses

1. The paper only reports performance improvements on RTX 5090. It is unclear how broadly applicable the method is to other hardware platforms.
2. The proposed method requires additional fine-tuning, whereas the compared baselines are training-free quantization methods. This difference in optimization setting may affect the fairness of the comparison.

---

> ### Author Rebuttal · Authors · 2026-03-31
>
> We thank reviewer K66S for the positive feedback and for recognizing our analysis of precision mismatch in attention QAT, as well as the practical value of our implementation and empirical results.
>
> We address the concerns and questions below.
>
> > W1 Other Hardware Platforms
>
> We implement a preliminary version of Attn-QAT on B200 and observe up to 1.39× speedup over BF16 FA4 [1]. The current gain is modest for two main reasons:
>
> 1. Kernel maturity.
> Kernel optimization on next-generation hardware is non-trivial, and **we expect further improvements as we further optimize the kernel**.
>
> 2. Hardware imbalance (compute vs. SFU throughput).
>
> B200 doubles GEMM/MMA throughput relative to H100 [2], but softmax (exp/SFU) throughput remains unchanged. As a result, **attention becomes softmax-bound, limiting the benefit of accelerating QK/PV GEMMs with NVFP4**.
>
> Looking forward, we expect larger gains on future hardware:
>
> B300 introduces ~2× higher exp throughput [2], which directly alleviates the current softmax bottleneck and should make GEMM acceleration more impactful.Rubin GPUs further improve execution scaling and provide higher effective throughput for FP16 exp operations [3]. This opens up a promising direction: combining NVFP4 QK/PV GEMMs with FP16 softmax, which may offer a better stability–performance trade-off for QAT.
>
> Overall, we view the current results as a lower bound, and anticipate more substantial speedups as both kernels and hardware evolve.
>
>
>
>  | Config                         | FP4 (ms) | FP4 TFLOPS | BF16 (ms) | BF16 TFLOPS | Speedup |
> |--------------------------------|----------|------------|-----------|-------------|---------|
> | b=1 s=256 h=16 d=128           | 0.015    | 37         | 0.015     | 35          | 1.01x   |
> | b=1 s=1024 h=16 d=128          | 0.023    | 379        | 0.025     | 338         | 1.12x   |
> | b=4 s=4096 h=16 d=128          | 0.336    | 1637       | 0.389     | 1413        | 1.16x   |
> | b=4 s=8192 h=16 d=128          | 1.259    | 1747       | 1.511     | 1455        | 1.20x   |
> | b=2 s=16384 h=16 d=128         | 2.467    | 1782       | 3.003     | 1464        | 1.22x   |
> | b=1 s=32768 h=16 d=128         | 4.884    | 1801       | 6.771     | 1299        | 1.39x   |
> | b=4 s=4096 h=32 d=128          | 0.655    | 1678       | 0.775     | 1418        | 1.18x   |
> | b=4 s=8192 h=32 d=128          | 2.501    | 1759       | 3.027     | 1453        | 1.21x   |
> | b=1 s=4096 h=12 d=128          | 0.104    | 986        | 0.117     | 882         | 1.12x   |
> | b=1 s=32768 h=12 d=128         | 3.856    | 1711       | 5.056     | 1305        | 1.31x   |
> | b=1 s=4096 h=24 d=128          | 0.152    | 1352       | 0.172     | 1198        | 1.13x   |
> | b=1 s=32768 h=24 d=128         | 7.551    | 1747       | 10.061    | 1311        | 1.33x   |
> | b=1 s=32768 h=24 d=64          | 7.170    | 920        | 7.284     | 906         | 1.02x   |
>
>
> >  W2 Additional Finetuning
>
> We agree the optimization settings differ, but this is unavoidable since no prior work explores 4-bit QAT for attention, so no fine-tuning baseline exists. Instead, we provide ablations (Table 2, Exp. 5–8) as controlled baselines, including a variant using SageAttention 3 as the forward kernel.
>
> Moreover, existing training-free ultra-low-bit methods are not directly comparable for LLMs: SageAttention 3 [4] has no validated LLM results, BitNet b1.58 [5] is only shown on 2B models for linear layers (vs. our 70B on attention), and QuIP [6] incurs ~50% inference overhead. Finally, many of these methods rely on software quantization and cannot leverage native FP4/FP8 Tensor Core support, making the comparison setting fundamentally different.
>
>
> References:
>
> [1] Flashattention-4: Algorithm and kernel pipelining co-design for asymmetric hardware scaling
>
> [2] https://developer.nvidia.com/blog/inside-nvidia-blackwell-ultra-the-chip-powering-the-ai-factory-era/#nvidia_gpu_chip_summary_comparison
>
> [3] https://developer.nvidia.com/blog/inside-the-nvidia-rubin-platform-six-new-chips-one-ai-supercomputer/
>
> [4] SageAttention3: Microscaling FP4 Attention for Inference and An Exploration of 8-bit Training
>
> [5] BitNet b1.58 2B4T Technical Report
>
> [6] QuIP: 2-Bit Quantization of Large Language Models With Guarantees

---

> > ### Author Rebuttal · Reviewer_K66S · 2026-04-03
> >
> > I will keep the positive score.

---

> > > ### Author Response · Authors · 2026-04-07
> > >
> > > We apologize for only addressing on the weaknesses section in our previous rebuttal. Below, we address the reviewer’s questions raised in the question section.
> > >
> > > > Q1 High-precision O
> > >
> > > Section 2.2 provides the theoretical basis for QAT. In Section 2.3, we show that **FlashAttention implicitly assumes identity between forward and backward precision**, which leads to incorrect gradient under quantization aware training. Using a high-precision O restores gradient correctness, and our empirical results validate this analysis.
> > >
> > > > Q2 Low-precision P
> > >
> > > Using low-precision P is the **correct formulation under QAT**, as it matches the forward-pass precision. While this does not significantly improve final accuracy, it leads to **more stable training**, evidenced by reduced gradient norms in Fig. 3(a). This stability benefit motivates the design choice.
> > >
> > > > Q3 Speedup breakdown
> > >
> > > All the observed speedup comes from  **removing preprocessing overhead** (e.g., Q/K smoothing). While simplifying P quantization reduces instruction complexity, we do not observe additional measurable kernel-level gains from this factor alone. We will clarify this attribution in the revision.
> > >
> > > Please let us know if any other concerns remain!

---

### Decision · Program_Chairs · 2026-04-30

**Decision:**

Accept (regular)

**Comment:**

This paper provides a systematic study of quantization-aware training (QAT) for FP4 attention, showing that naively applying QAT to FlashAttention-style implementations leads to training instability, primarily due to precision mismatches (the heavy-tailed distribution of attention activations and the limited dynamic range of FP4) between the forward and backward passes. To address this issue, the authors proposes Attn-QAT, which recomputes the attention scores in low precision during the backward pass to match the forward computation, and additionally computes and stores the attention output in high precision to enable stable gradient computation. Additionally, the authors implement custom Triton kernels for training and CUDA kernels for inference. Experiments with multiple diffusion models and LLMs on different tasks to show the efficacy of Attn-QAT.

The paper was initially/finally scored (4,5,4,5)/(4,5,5,5) by four reviewers, who mostly recognized the importance of the problem, the strong motivation, the insightful observations, the novel idea and the promising performance, and raised several concerns about 1) the theoretical contribution is not significant, lacking theoretical explanations for key observations; 2) evaluation with more HW platforms; 2) the method requires fine-tuning, leading to extra overhead; 3) lack a clear analysis of how QAT affects the distribution of attention activations or mitigates the impact of outliers; 4) some ablations are missing; 5) somewhat narrow research scope.

The authors provided detailed responses to address these concerns. All reviewers acknowledged the rebuttal. Finally, three reviewers (5hyu, Qcve, 3hrY) were fully satisfied with the rebuttal and consistently gave the positive score Accept, and the weakly-positive Reviewer K66S would be also satisfied with further responses from the authors. The AC read the paper, the reviews, the rebuttal and the reviewers' feedback. I agree with reviewers that this paper has a strong motivation, insightful observations and good novelty, and shows good performance with practical values, and thus recommend an “Accept”. The authors are encouraged to include additional experiments, discussions and clarifications in the final version of paper.